# Provably Personalized and Robust Federated Learning

**Mariel Werner**
*Department of Electrical Engineering and Computer Sciences,*
*University of California, Berkeley*

**Lie He**
*Machine Learning and Optimization Laboratory (MLO),*
*EPFL, Switzerland*

**Michael Jordan**
*Department of Electrical Engineering and Computer Sciences,*
*University of California, Berkeley*

**Martin Jaggi**
*Machine Learning and Optimization Laboratory (MLO),*
*EPFL, Switzerland*

**Sai Praneeth Karimireddy**
*Department of Electrical Engineering and Computer Sciences,*
*University of California, Berkeley*

**Reviewed on OpenReview:** *https://openreview.net/forum?id=B0uBSSUyOG*

## Abstract

Clustering clients with similar objectives and learning a model per cluster is an intuitive and interpretable approach to personalization in federated learning. However, doing so with provable and optimal guarantees has remained an open challenge. In this work, we formalize personalized federated learning as a stochastic optimization problem. We propose simple clustering-based algorithms which iteratively identify and train within clusters, using local client gradients. Our algorithms have optimal convergence rates which asymptotically match those obtained if we knew the true underlying clustering of the clients, and are provably robust in the Byzantine setting where some fraction of the clients are malicious.

## 1 Introduction

We consider the federated learning setting in which there are $N$ clients with individual loss functions $\{f_i\}_{i \in [N]}$ who seek to jointly train a model or multiple models. The defacto algorithm for problems in this setting is FedAvg (McMahan et al., 2017) which has an objective of the form

$$x^*_{\text{FedAvg}} = \arg\min_{x \in \mathcal{X}} \frac{1}{N} \sum_{i \in [N]} f_i(x). \tag{1}$$

From (1), we see that FedAvg optimizes the average of the client losses. In many real-world cases however, clients' data distributions are heterogeneous, making such an approach unsuitable since the global optimum (1) may be very far from the optima of individual clients. Rather, we want algorithms which identify clusters of the clients that have relevant data for each other and that only perform training within each cluster. However, this is a challenging exercise since 1) it is unclear what it means for data distributions of two clients to be useful for each other, or 2) how to automatically identify such subsets without expensive multiple retraining (Zamir et al., 2018). In this work we propose algorithms which iteratively and simultaneously 1) identify $K$ clusters amongst the clients by clustering their gradients and 2) optimize the clients' losses within each cluster.

## 1.1 Related Work

**Personalization via Clustering.** Personalization in federated learning has recently enjoyed tremendous attention (see Tan et al. (2022); Kulkarni et al. (2020) for surveys). We focus on gradient-based clustering methods for personalized federated learning. Several recent works propose and analyze clustering methods. Sattler et al. (2021) alternately train a global model with FedAvg and partition clients into smaller clusters based on the global model's performance on their local data. Mansour et al. (2021) and Ghosh et al. (2020) instead train personalized models from the start (as we do) without maintaining a global model. They iteratively update $K$ models and, using empirical risk minimization, assign each of $N$ clients one of the models at every step. In Section 2 we analyze these algorithms on constructed examples and in Section 3.1.3 compare them to our method.

Since our work is closest to Ghosh et al. (2020), we highlight key similarities and differences. **Similarities**: 1) We both design stochastic gradient descent- and clustering-based algorithms for personalized federated learning. 2) We both assume sufficient intra-cluster closeness and inter-cluster separation of clients for the clustering task (Assumptions 1 and 2 in their work; Assumptions 4 and 5 in ours). 3) Our convergence rates both scale inversely with the number of clients and the inter-cluster separation parameter $\Delta$. **Differences**: 1) They assume strong convexity of the clients' loss objectives, while our guarantees hold for all smooth (convex and non-convex) functions. 2) They cluster clients based on similarity of loss-function values whereas we cluster clients based on similarity of gradients. We show that clustering based on loss-function values instead of gradients can be overly sensitive to model initialization (see Fig. 1b). 3) Since we determine clusters based on distances in gradient space, we are able to apply an aggregation rule which makes our algorithm robust to some fraction of malicious clients. They determine cluster identity based on loss-function value and do not provide robustness guarantees.

Recently, Even et al. (2022) established lower bounds showing that the optimal strategy is to cluster clients who share the same optimum. Our algorithms and theoretical analysis are inspired by this lower-bound, and our gradient-based clustering approach makes our algorithms amenable to analysis à la their framework.

**Multitask learning.** Our work is closely related to multitask learning, which simultaneously trains separate models for different-but-related tasks. Smith et al. (2017) and Li et al. (2021) both cast personalized federated learning as a multitask learning problem. In the first, the per-task models jointly minimize an objective that encodes relationships between the tasks. In the second, models are trained locally (for personalization) but regularized to be close to an optimal global model (for task-relatedness). These settings are quite similar to our setting. However, we use assumptions on gradient (dis)similarity across the domain space to encode relationships between tasks, and we do not maintain a global model.

**Robustness.** Our methods are provably robust in the Byzantine (Lamport et al., 2019; Blanchard et al., 2017) setting, where clients can make arbitrary updates to their gradients to corrupt the training process. Several works on Byzantine robust distributed optimization (Blanchard et al., 2017; Yin et al., 2018; Damaskinos et al., 2018; Guerraoui & Rouault, 2018; Pillutla et al., 2022) propose aggregation rules in lieu of averaging as a step towards robustness. However, Baruch et al. (2019); Xie et al. (2020) show that these rules are not in fact robust and perform poorly in practice. Karimireddy et al. (2021) are the first to provide a provably Byzantine-robust distributed optimization framework by combining a novel aggregation rule with momentum-based stochastic gradient descent. We use a version of their centered-clipping aggregation rule to update client gradients. Due to this overlap in aggregation rule, components of our convergence results are similar to their Theorem 6. However, our analysis is significantly complicated by our personalization and clustering structure. In particular, all non-malicious clients in Karimireddy et al. (2021; 2022) have the same optimum and therefore can be viewed as comprising a single cluster, whereas we consider multiple clusters of clients (without necessarily assuming clients are i.i.d. within a cluster). The personalization algorithm in Li et al. (2021) also has robustness properties, but they are only demonstrated empirically and analyzed on toy examples.

**Recent Empirical Approaches.** Two recent works (Wang et al., 2022; Wu et al., 2023) examine the setting in which clients' marginal distributions $p(x)$ differ, whereas most prior work only allows their conditional distributions $p(y|x)$ to differ. One of our experiments (Section 4.1) assumes heterogeneity between clients'

marginal distributions, while the others (Section 4.2) assume heterogeneity only between their conditional distributions. In Wang et al. (2022), the server maintains a global pool of modules (neural networks) from which clients, via a routing algorithm, efficiently select and combine sub-modules to create personalized models that perform well on their individual distributions. Extending the work of Marfoq et al. (2021), Wu et al. (2023) model each client's joint distribution as a mixture of Gaussian distributions, with the weights of the mixture personalized to each client. They then propose a federated Expectation-Maximization algorithm to optimize the parameters of the mixture model. In general, the contributions and style of our work and these others differ significantly. We focus on achieving and proving optimal theoretical convergence rates which we verify empirically, whereas Wang et al. (2022) and Wu et al. (2023) emphasize empirical application over theoretical analysis.

## 1.2 Our Contributions

To address the shortcomings in current approaches, we propose two personalized federated learning algorithms, which simultaneously cluster similar clients and optimize their loss objectives in a personalized manner. In each round of the procedure, we examine the client gradients to identify the cluster structure as well as to update the model parameters. Importantly, ours is the first method with theoretical guarantees for general non-convex loss functions, and not just restrictive toy settings. We show that our method enjoys both nearly optimal convergence, while also being robust to some malicious (Byzantine) client updates. This is again the first theoretical proof of the utility of personalization for Byzantine robustness. Specifically in this work,

- We show that existing or naive clustering methods for personalized learning, with stronger assumptions than ours, can fail in simple settings (Fig. 1).

- We design a robust clustering subroutine (Algorithm 3) whose performance improves with the separation between the cluster means and the number of data points being clustered. We prove nearly matching lower bounds showing its near-optimality (Theorem 2), and we show that the error due to malicious clients scales smoothly with the fraction of such clients (Theorem 1).

- We propose two personalized learning algorithms (Algorithm 2 and Algorithm 4) which converge at the optimal $\mathcal{O}(1/\sqrt{n_i T})$ rate in $T$ for stochastic gradient descent for smooth non-convex functions and linearly scale with $n_i$, the number of clients in client $i$'s cluster.

- We empirically verify our theoretical assumptions and demonstrate experimentally that our learning algorithms benefit from collaboration, scale with the number of collaborators, are competitive with SOTA personalized federated learning algorithms, and are not sensitive to the model's initial weights (Section 4).

## 2 Existing Clustering Methods for Personalized Federated Learning

Our task at hand in this work is to simultaneously learn the clustering structure amongst clients and minimize their losses. Current methods do not rigorously check similarity of clients throughout the training process. Therefore they are not able to correct for early-on erroneous clustering (e.g. due to gradient stochasticity, model initialization, or the form of loss-functions far from their optima). In the next section we demonstrate such failure modes of existing algorithms.

### 2.1 Failure Modes of Existing Methods

The first algorithm we discuss, Myopic-Clustering, does not appear in the existing literature, but we create it in order to motivate the design of our method (Algorithm 2). In particular, it is a natural first step towards our method, but has limitations which we correct when designing our algorithm.

**Myopic-Clustering (Algorithm 1).** At every step, each client computes their gradient at their current model and sends the gradient to a central server. The server clusters the gradients and sends each cluster

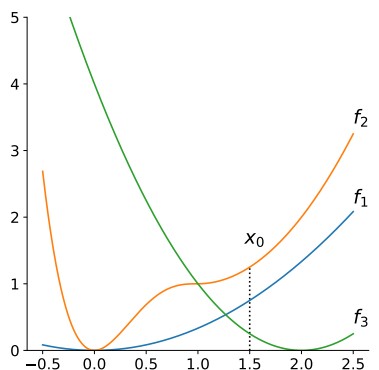 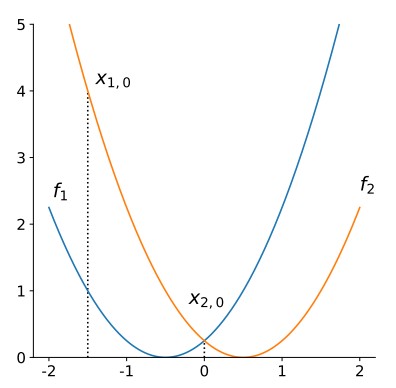 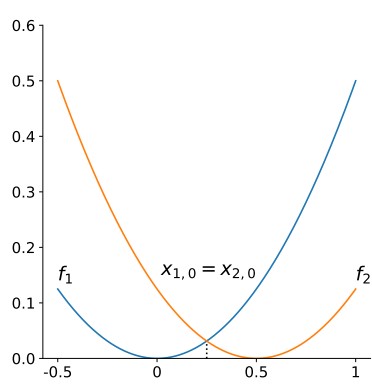

(a) **Myopic-Clustering** ($\eta = 0.5$). **Correct clustering**: $\{1,2\}$ and $\{3\}$. Client $\{2\}$ gets stuck at $x = 1$, not reaching its optimum, and clients $\{1,3\}$ converge to their optima. All gradients being 0 at this point, the clients are incorrectly clustered together: $\{1,2,3\}$.

(b) **IFCA/HypCluster**. **Correct clustering**: $\{1\},\{2\}$. Both clients' function values are smaller at initialization point $x_{2,0}$ than $x_{1,0}$ causing IFCA/HypCluster to initially cluster them together. Since the average of the clients' gradients at $x_{2,0}$ is 0, the models never update and the algorithm thinks the initial erroneous clustering, $\{1,2\}$ is correct.

(c) **Clustered Federated Learning**. **Correct clustering**: $\{1\}$, $\{2,3\}$ (client 3 not drawn due to its stochastic gradient – details on pg. 5). Clustered FL averages gradients of clients $\{1\}$ and $\{2\}$ to 0, clustering them together, and with non-0 probability clusters $\{3\}$ separately due to its stochastic gradient. Based on this initial erroneous, the algorithm partitions the clients $\{1,2\}$ and $\{3\}$ and recursively runs on each group, never recovering the correct clustering.

Figure 1: We show how existing personalized FL algorithms miscluster and fail to converge on constructed examples.

center to the clients assigned to that cluster. Each client then performs a gradient descent update on their model with their received cluster center. This is a natural federated clustering procedure and it is communication-efficient ($\mathcal{O}(N)$). However, it has two issues: 1) If it makes a clustering mistake at one step, models will be updated with the wrong set of clients. This can cause models to diverge from their optima, gradients of clients in the same cluster to drift apart, and gradients of clients in different clusters to drift together, thus obscuring the correct clustering going forward. Furthermore, these errors can compound over rounds. 2) Even if Myopic-Clustering clusters clients perfectly at each step, the clients' gradients will approach zero as the models converge to their optima. This means that clients from different clusters will appear to belong to the same cluster as the algorithm converges and all clients will collapse into a single cluster. The following example (Fig. 1a) demonstrates these failure modes of Myopic-Clustering.

Let $N = 3$ and $K = 2$, with client loss functions

$$f_1(x) = \frac{1}{6\eta}x^2$$

$$f_2(x) = \begin{cases} 4(x-1)^3 + 3(x-1)^4 + 1 & x < 1 \\ \frac{1}{2\eta}(x-1)^2 + 1 & x \geq 1, \end{cases}$$

$$f_3(x) = \frac{1}{2\eta}(x-2)^2,$$

where $\eta$ is the learning rate of the algorithm. With this structure, clients $\{1,2\}$ share the same global minimum and belong to the same cluster, and client $\{3\}$ belongs to its own cluster. Suppose Myopic-Clustering is initialized at $x_0 = 1.5$. At step 1, the client gradients computed at $x_0 = 1.5$ are $1/2\eta$, $1/2\eta$, and $-1/2\eta$ respectively. Therefore, clients $\{1,2\}$ are correctly clustered together and client $\{3\}$ alone at this step.

---

**Algorithm 1** Myopic-Clustering

---

**Input** Learning rate: $\eta$. Initial parameters: $\{x_{1,0} = ... = x_{N,0} = x_0\}$.

1: **for** round $t \in [T]$ **do**
2:      **for** client $i$ in [N] **do**
3:          Client $i$ sends $g_i(x_{i,t-1})$ to server.
4:          Server clusters $\{g_i(x_{i,t-1})\}_{i \in [N]}$, generating cluster centers $\{v_{k,t}\}_{k \in [K]}$.
5:          Server sends $v_{k_i,t}$ to client $i$, where $k_i$ denotes the cluster to which client $i$ is assigned.
6:          Client $i$ computes update: $x_{i,t} = x_{i,t-1} - \eta v_{k_i,t}$.
7: **Output:** Personalized parameters: $\{x_{1,T}, ..., x_{N,T}\}$.

---

After updates, the clients' parameters will next be $x_{1,1} = 1, x_{2,1} = 1, x_{3,1} = 2$ respectively. At this point, clients $\{2, 3\}$ will be incorrectly clustered together since their gradients will both be 0, while client $\{1\}$ will be clustered alone. As the algorithm proceeds, clients $\{2, 3\}$ will always be clustered together and will remain at $x = 1$ and $x = 2$ respectively, while client $\{1\}$ will converge to its optimum at $x = 0$. Consequently, two undesirable things happen: 1) Client $\{2\}$ gets stuck at the saddle point at $x = 1$ which occurred when it was incorrectly clustered with client $\{3\}$ at $t = 1$ and subsequently did not recover. 2) All gradients converge to 0, so at the end of the algorithm all clients are clustered together.

To further motivate the design choice for our algorithms, we now discuss three clustering-based algorithms in the literature on personalized federated learning. In particular, we generate counter-examples on which they fail and show how our algorithm avoids such pitfalls.

The first two algorithms IFCA (Ghosh et al., 2020) and HypCluster (Mansour et al., 2021) are closely related. They both cluster loss function values rather than gradients, and like our algorithm they avoid the myopic nature of Myopic-Clustering by, at each step, computing all client losses at all current cluster parameters to determine the clustering. However, as we show in the next example (Fig. 1b), they are brittle and sensitive to initialization.

**IFCA (Ghosh et al., 2020).** Let $N = 2$, $K = 2$ with loss functions

$$f_1(x) = (x + 0.5)^2$$
$$f_2(x) = (x - 0.5)^2,$$

and initialize clusters 1 and 2 at $x_{1,0} = -1.5$ and $x_{2,0} = 0$ respectively. Given this setup, both clients initially select cluster 2 since their losses at $x_{2,0}$ are smaller than at $x_{1,0}$.

Option I: At $x_{2,0} = 0$, the client gradients will average to 0. Consequently the models will remain stuck at their initializations, and both clients will be incorrectly assigned to cluster 2.

Option II: Both clients individually run $\tau$ steps of gradient descent starting at their selected model $x_{2,0}$ (i.e. perform the `LocalUpdate` function in line 18 of IFCA). Since the clients' individually updated models will be symmetric around 0 after this process, the server will compute cluster 2's model update in line 15 of IFCA as: $x_{2,1} \leftarrow 0 = x_{2,0}$. Consequently, the outcome is the same as in Option I: the models never update and both clients are incorrectly assigned to cluster 2.

**HypCluster (Mansour et al., 2021).** This algorithm is a centralized version of Option II of IFCA. The server alternately clusters clients by loss function value and runs stochastic gradient descent per-cluster using the clients' data. It performs as Option II of IFCA on the example above.

Finally, we discuss Clustered Federated Learning, the algorithm proposed in Sattler et al. (2021), which runs the risk of clustering too finely, as in the next example (Fig. 1c).

**Clustered Federated Learning (Sattler et al., 2021).** Clustered Federated Learning operates by recursively bi-partitioning the set of clients based on the clients' gradient values at the FedAvg optimum.

Consider the following example. Let $N = 3$ and $K = 2$ with client gradients

$$g_1(x) = x$$
$$g_2(x) = x - 1/2$$
$$g_3(x) = \begin{cases} x & \text{w.p. } 1/2 \\ x - 1 & \text{w.p. } 1/2. \end{cases}$$

Therefore the correct clustering here is $\{1\}$ and $\{2, 3\}$. The FedAvg optimum is $x^*_{\text{FedAvg}} = 1/4$, at which the clients' gradient values are $g_1(1/4) = 1/4$, $g_2(1/4) = -1/4$ and $g_3 = 1/4$ w.p. $1/2$. Based on this computation, Clustered Federated Learning partitions the client set into $\{1, 2\}$ and $\{3\}$ w.p. $1/2$ and then proceeds to run the algorithm separately on each sub-cluster. Therefore, the algorithm never corrects its initial error in separating clients $\{2\}$ and $\{3\}$.

The behaviour of these algorithms motivates our method Federated-Clustering, which by rigorously checking client similarity at every step of the training process can recover from past clustering errors.

## 3 Proposed Method: Federated-Clustering (Algorithm 2)

At a high level, Federated-Clustering works as follows. Each client $i$ maintains a personalized model which, at every step, it broadcasts to the other clients $j \neq i$. Then each client $j$ computes its gradient on clients $i$'s model parameters and sends the gradient to client $i$. Finally, client $i$ runs a clustering procedure on the received gradients, determines which other clients have gradients closest to its own at its current model, and updates its current model by averaging the gradients of these similar clients. By the end of the algorithm, ideally each client has a model which has been trained only on the data of similar clients.

The core of Federated-Clustering is a clustering procedure, Threshold-Clustering (Algorithm 3), which identifies clients with similar gradients at each step. This clustering procedure, which we discuss in the next section, has two important properties: it is robust and its error rate is near-optimal.

**Notation.** For an arbitrary integer $N$, we let $[N] = \{1, ..., N\}$. We take $a \gtrsim b$ to mean there is a sufficiently large constant $c$ such that $a \geq cb$, $a \lesssim b$ to mean there is a sufficiently small constant $c$ such that $a \leq cb$, and $a \approx b$ to mean there is a constant $c$ such that $a = cb$. We write $i \sim j$ if clients $i$ and $j$ belong to the same cluster, $i \overset{i.i.d.}{\sim} j$ if they belong to the same cluster and their data is drawn independently from identical distributions (we will sometimes equivalently write $z_i \overset{i.i.d.}{\sim} z_j$, where $z_i$ and $z_j$ are arbitrary points drawn from clients $i$'s and $j$'s distributions), and $i \nsim j$ if they belong to different clusters. For two different clients $i$ and $j$, same cluster or not, we write $i \neq j$. Finally, $n_i$ denotes the number of clients in client $i$'s cluster, $\delta_i = n_i/N$ denotes the fraction of clients in client $i$'s cluster, and $\beta_i$ denotes the fraction of clients that are malicious from client $i$'s perspective.

### 3.1 Analysis of Clustering Procedure

Given the task of clustering $N$ points into $K$ clusters, at step $l$ our clustering procedure has current estimates of the $K$ cluster-centers, $v_{1,l}, ..., v_{K,l}$. To update each estimate $v_{k,l+1} \leftarrow v_{k,l}$, it constructs a ball of radius $\tau_{k,l}$ around $v_{k,l}$. If a point falls inside the ball, the point retains its value; if it falls outside the ball, its value is mapped to the current cluster-center estimate. The values of all the points are then averaged to set $v_{k,l+1}$ (update rule (4)). The advantage of this rule is that it is very conservative. If our algorithm is confident that its current cluster-center estimate is close to the true cluster mean (i.e. there are many points nearby), it will confidently improve its estimate by taking a large step in the right direction (where the step size and direction are determined mainly by the nearby points). If our algorithm is not confident about being close to the cluster mean, it will tentatively improve its estimate by taking a small step in the right direction (where the step size and direction are small since the majority of points are far away and thus do not change the current estimate).

To analyze the theoretical properties of this procedure, we look at a natural setting in which clients within the same cluster have i.i.d. data (for analysis of our federated learning algorithm, we will relax this strong

notion of intra-cluster similarity). In particular, in our setting there are $N$ points $\{z_1, ..., z_N\}$ which can be partitioned into $K$ clusters within which points are i.i.d.. We assume the following.

- **Assumption 1** (Intra-cluster Similarity): For all $i \sim j$,

$$z_i \overset{i.i.d.}{\sim} z_j.$$

- **Assumption 2** (Inter-cluster Separation): For all $i \nsim j$,

$$\|\mathbb{E}z_i - \mathbb{E}z_j\|^2 \geq \Delta^2.$$

- **Assumption 3** (Bounded Variance): For all $z_i$,

$$\mathbb{E}\|z_i - \mathbb{E}z_i\|^2 \leq \sigma^2.$$

**Theorem 1.** *Suppose there $N$ points $\{z_i\}_{i \in [N]}$ for which Assumptions [1-3] hold with inter-cluster separation parameter $\Delta \gtrsim \sigma/\delta_i$. Running Algorithm 3 for*

$$l \gtrsim \max\left\{1, \max_{i \in [N]} \frac{\log(\sigma/\Delta)}{\log(1 - \delta_i/2)}\right\}$$

*steps with fraction of malicious clients $\beta_i \lesssim \delta_i$ and thresholding radius $\tau \approx \sqrt{\delta_i \sigma \Delta}$ guarantees that*

$$\mathbb{E}\|v_{k_i,l} - \mathbb{E}z_i\|^2 \lesssim \frac{\sigma^2}{n_i} + \frac{\sigma^3}{\Delta} + \beta_i \sigma \Delta. \tag{2}$$

*Proof.* See A.1. □

Supposing $\beta_i = 0$, if we knew the identity of all points within $z_i$'s cluster, we would simply take their mean as the cluster-center estimate, incurring estimation error of $\sigma^2/n_i$ (i.e. the sample-mean's variance). Since we don't know the identity of points within clusters, the additional factor of $\sigma^3/\Delta$ in (2) is the price we pay to learn the clusters. This additional term scales with the difficulty of the clustering problem. If true clusters are well-separated and/or the variance of the points within each cluster is small (i.e. $\Delta$ is large, $\sigma^2$ is small), then the clustering problem is easier and our bound is tighter. If clusters are less-well-separated and/or the variance of the points within each cluster is large, accurate clustering is more difficult and our bound weakens.

**Setting $\tau$.** To achieve the rate in (2), we set $\tau \approx \sqrt{\sigma \Delta}$, which is the geometric mean of the standard deviation, $\sigma$, of points belonging to the same cluster and the distance, $\Delta$, to a different cluster. The intuition for this choice is that we want the radius for each cluster to be at least as large as the standard deviation of the points belonging to that cluster in order to capture in-cluster points. The radius could be significantly larger than the standard deviation if $\Delta$ is large, thus capturing many non-cluster points as well. However, the conservative nature of our update rule (4) offsets this risk. By only updating the center with a step-size proportional to the *fraction* of points inside the ball, it limits the influence of any mistakenly captured points.

Threshold-Clustering has two important properties which we now discuss: it is Byzantine robust and has a near-optimal error rate.

### 3.1.1 Robustness

We construct the following definition to characterize the robustness of Algorithm 3.

**Definition 1** (Robustness). *An algorithm $\mathcal{A}$ is robust if the error introduced by bad clients can be bounded i.e. malicious clients do not have an arbitrarily large effect on the convergence. Specifically, for a specific objective, let $\mathcal{E}_1$ be the base error of $\mathcal{A}$ with no bad clients, let $\beta$ be the fraction of bad clients, and let $\mathcal{E}_2$ be some bounded error added by the bad points. Then $\mathcal{A}$ is robust if*

$$Err(\mathcal{A}) \leq \mathcal{E}_1 + \beta \mathcal{E}_2.$$

**Threat model.** Our clustering procedure first estimates the centers of the $K$ clusters from the $N$ points and constructs a ball of radius $\tau_k$ around the estimated center of each cluster $k$. If a point falls inside the ball, the point retains its value; if it falls outside the ball, its value is mapped to the current cluster-center estimate. Following the update rule (4), the values of all the points are averaged to update the cluster-center estimate. Therefore, a bad point that wants to distort the estimate of the $k$'th cluster's center has the most influence by placing itself just within the boundary of the ball around that cluster-center, i.e. at $\tau_k$-distance from the cluster-center.

From (2) we see that the base squared-error of Algorithm 3 in estimating $z_i$'s cluster-center is $\lesssim \sigma^2/n_i + \sigma^3/\Delta$, and that the bad points introduce extra squared-error of order $\sigma\Delta$. Given our threat model, this is exactly expected. The radius around $z_i$'s cluster-center is order $\sqrt{\sigma\Delta}$. Therefore, bad points placing themselves at the edge of the ball around $z_i$'s cluster-center estimate will be able to distort the estimate by order $\sigma\Delta$. The scaling of this extra error by $\beta_i$ satisfies our definition of robustness, and the error smoothly vanishes as $\beta_i \to 0$.

### 3.1.2 Near-Optimality

The next result shows that the upper bound (2) on the estimation error of Algorithm 3 nearly matches the best-achievable lower bound. In particular, it is tight within a factor of $\sigma/\Delta$.

**Theorem 2** (Near-optimality of **Threshold-Clustering**). *For any algorithm $\mathcal{A}$, there exists a mixture of distributions $\mathcal{D}_1 = (\mu_1, \sigma^2)$ and $\mathcal{D}_2 = (\mu_2, \sigma^2)$ with $\|\mu_1 - \mu_2\| \geq \Delta$ such that the estimator $\hat{\mu}_1$ produced by $\mathcal{A}$ has error*

$$\mathbb{E}\|\hat{\mu}_1 - \mu_1\|^2 \geq \Omega\left(\frac{\sigma^4}{\Delta^2} + \frac{\sigma^2}{n_i}\right).$$

*Proof.* See A.2. □

### 3.1.3 Federated-Clustering on Examples in Section 2.1

We describe how Federated-Clustering successfully handles the examples in Section 2.1.

**Example 1: Fig. 1a.** Federated-Clustering checks at every step the gradient values of all $N$ clients at the current parameters of all $K$ clusters. This verification process avoids the type of errors made by Myopic-Clustering. For instance, at $t = 1$ when Myopic-Clustering makes its error, Federated-Clustering computes the gradients of all clients at client $\{1\}$'s current parameters: $g_1(1) = 1/3\eta$, $g_2(1) = 0$, and $g_3(1) = -1/\eta$. Therefore it correctly clusters $\{1, 2\}$ together at this point, and client $\{2\}$'s parameters update beyond the saddle-point and converge to the global minimum at $x = 0$.

**Example 2: Fig. 1b.** By clustering clients based on gradient instead of loss value, Federated-Clustering initially computes the clients' gradients of $+1$ and $-1$ respectively at $x_{2,0} = 0$, and given the continued separation of their gradients around 0 as the algorithm converges, correctly identifies that they belong to different clusters.

**Example 3: Fig. 1c.** Recall how Clustered FL fails on this example. Based on an initial clustering error, it partitions the clients incorrectly early on and then evaluates each subset separately going forward, thus never recovering the correct clustering. Our algorithm avoids this type of mistake by considering all clients during each clustering at every step.

### 3.2 Analysis of Federated-Clustering

We now proceed with the analysis of Federated-Clustering. First, we establish necessary assumptions: intra-cluster similarity, inter-cluster separation, bounded variance of stochastic gradients, and smoothness of loss objectives.

- **Assumption 4** (Intra-cluster Similarity): For all $x$, $i \sim j$, and some constant $A \geq 0$,

$$\|\nabla f_i(x) - \nabla \bar{f}_i(x)\|^2 \leq A^2 \|\nabla \bar{f}_i(x)\|^2,$$

where $\bar{f}_i(x) \triangleq \frac{1}{n_i} \sum_{j \sim i} f_j(x)$.

- **Assumption 5** (Inter-cluster Separation): For all $x$, $i \not\sim j$, and some constants $\Delta$, $D \geq 0$,

$$\|\nabla f_i(x) - \nabla f_j(x)\|^2 \geq \Delta^2 - D^2 \|\nabla f_i(x)\|^2.$$

This formulation is motivated by the information theoretic lower-bounds of Even et al. (2022) who show that the optimal clustering strategy is to group all clients with the same optimum (even if they are non-iid). Assumptions 4 and 5 are in fact a slight strengthening of this very statement. To see this, note that for a client with loss function $f_i(x)$, belonging to cluster $\bar{f}_i(x)$ with a first-order stationary points $\bar{x}^*$, Assumption 4 implies that if $\nabla \bar{f}_i(\bar{x}^*) = 0 \Rightarrow \nabla f_i(\bar{x}^*) = 0$, and so $\bar{x}^*$ is also a stationary point for client $i$. Thus, all clients within a cluster have shared stationary points. Assumption 4 further implies that the gradient difference elsewhere away from the optima is also bounded. This latter strengthening is motivated by the fact the the loss functions are smooth, and so the gradients cannot diverge arbitrarily as we move away from the shared optima. In fact, it is closely related to the strong growth condition (equation (1) in Vaswani et al. (2019)), which is shown to be a very useful notion in practical deep learning. We also empirically verify its validity in Fig. 2 (left).

Similarly, Assumption 5 is a strengthening of the condition that clients across different clusters need to have different optima. For two clients $i$ and $j$ who belong to different clusters and with first-order stationary points $x_i^*$ and $x_j^*$, Assumption 5 implies that $\|\nabla f_i(x_j^*)\|^2 \geq \Delta$ and $\|\nabla f_j(x_i^*)\|^2 \geq \Delta$. Thus, they do not share any common optimum. Similar to the Assumption 4, Assumption 5 also describes what happens elsewhere away from the optima - it allows for the difference between the gradients to be smaller than $\Delta$ as we move away from the stationary points. Again, this specific formulation is motivated by smoothness of the loss function, and empirical validation (Fig. 2, right).

In Corollary 1, we give a setting and precise $A$, $D$, and $\Delta$ for which Assumptions 4 and 5 are satisfied.

- **Assumption 6** (Bounded Variance of Stochastic Gradients): For all $x$,

$$\mathbb{E}\|g_i(x) - \nabla f_i(x)\|^2 \leq \sigma^2,$$

where $\mathbb{E}[g_i(x)|x] = \nabla f_i(x)$ and each client $i$'s stochastic gradients $g_i(x)$ are independent.

- **Assumption 7** (Smoothness of Loss Functions): For any $x, y$,

$$\|\nabla f_i(x) - \nabla f_i(y)\| \leq L\|x - y\|.$$

**Theorem 3.** *Let* Assumptions [4-7] *hold with inter-cluster separation parameter* $\Delta \gtrsim \max(1,A^4)\max(1,D^2)\sigma/\delta_i$. *Under these conditions, suppose we run Algorithm 2 for $T$ rounds with learning rate $\eta \leq 1/L$, fraction of malicious clients $\beta_i \lesssim \delta_i$, and batch size $|B_i| \gtrsim \min(\sqrt{\max(1, A^2)(\sigma^2/n_i + \sigma^3/\Delta + \beta_i\sigma\Delta)m_i}, m_i)$, where $m_i$ is the size of client $i$'s training dataset. If, in each round $t \in [T]$, we cluster with radius $\tau \approx \sqrt{\delta_i\sigma\Delta}$ for*

$$l \geq \max \left\{ 1, \max_{i \in N]} \frac{\log(\sigma/\sqrt{|B_i|}\Delta)}{\log(1 - \delta_i/2)} \right\}$$

*steps, then*

$$\frac{1}{T} \sum_{t=1}^{T} \mathbb{E}\|\nabla f_i(x_{i,t-1})\|^2 \lesssim \sqrt{\frac{\max(1, A^2)(\sigma^2/n_i + \sigma^3/\Delta + \beta_i\sigma\Delta)}{T}}. \tag{3}$$

*Proof.* See A.3. $\qquad \square$

We note a few things. 1) The rate in (3) is the optimal rate in $T$ for stochastic gradient descent on non-convex functions (Arjevani et al., 2023). 2) The dependence on $\sqrt{\sigma^2/n_i}$ is intuitive, since convergence error should increase as the variance of points in the cluster increases and decrease as the number of points in the cluster increases. It is also optimal as shown in Even et al. (2022). 3) The dependence on $\sqrt{\beta_i \sigma \Delta}$ is also expected. We choose a radius $\tau \approx \sqrt{\sigma \Delta}$ for clustering. Given our threat model, the most adversarial behavior of the bad clients from client $i$'s perspective is to place themselves at the edge of the ball surrounding the estimated location of $i$'s gradient, thus adding error of order $\sqrt{\beta_i \sigma \Delta}$. When there are no malicious clients, this extra error vanishes. 4) If the constraint on batch-size in Theorem 3 requires $|B_i| = m_i$, then the variance of stochastic gradients vanishes and the standard $\mathcal{O}(1/T)$ rate for deterministic gradient descent is recovered (see equation (18) in proof). We also note that as long as there are no malicious clients (i.e. $\beta_i \sigma \Delta$ term is 0), there are a large enough number of clients $n_i$ in the cluster, and inter-cluster separation $\Delta$ is sufficiently larger than the inter-cluster-variance $\sigma^2$, then minimum-batch size will likely be less than $m_i$.

If losses are smooth and strongly convex, the following convergence rate is achievable.

**Corollary 1.** *Suppose client losses are $L$-smooth and $\mu$-strongly convex, and that clusters are defined by clients with the same optimum. Specifically, let $x_i^*$ be client $i$'s optimum. Define $\Delta = \max_{j \not\sim i} \frac{1}{\sqrt{2}} \|\nabla f_j(x_i^*)\|$ and assume losses are such that $\Delta \gtrsim L^6 \sigma / \mu^6 \delta_i$. If we run Algorithm 2 for $T$ rounds with learning rate $\eta \leq 1/L$, fraction of malicious clients $\beta_i \lesssim \delta_i$, batch size $|B_i| \gtrsim \min\left(\sqrt{(L/\mu)^2 (\sigma^2/n_i + \sigma^3/\Delta + \beta_i \sigma \Delta) m_i}, m_i\right)$, where $m_i$ is the size of client $i$'s training dataset, and cluster in each round $t \in [T]$ with radius $\tau \approx \sqrt{\delta_i \sigma \Delta}$ for*

$$l \geq \max\left\{1, \max_{i \in N]} \frac{\log(\sigma/\sqrt{|B_i|}\Delta)}{\log(1 - \delta_i/2)}\right\}$$

*steps, then*

$$\frac{1}{T} \sum_{t=1}^{T} \mathbb{E}\|\nabla f_i(x_{i,t-1})\|^2 \lesssim \sqrt{\frac{(L/\mu)^2 (\sigma^2/n_i + \sigma^3/\Delta + \beta_i \sigma \Delta)}{T}}.$$

*Proof.* See A.4. □

**Privacy.** Since Federated-Clustering requires clients to compute distances between gradients, they must share their models and gradients which compromises privacy. The focus of our work is not on optimizing privacy, so we accommodate only the lightest layer of privacy for federated learning: sharing of models and gradients rather than raw data. Applying more robust privacy techniques is a direction for future work. In the meantime, we refer the reader to the extensive literature on differential privacy, multi-party computation, and homomorphic encryption in federated learning.

**Communication Overhead.** At each step, Federated-Clustering requires $\mathcal{O}(N^2)$ rounds of communication since each client sends its model to every other client, evaluates its own gradient at every other client's model and then sends this gradient back to the client who owns the model. We pay this communication price to mitigate the effect of past clustering mistakes. For example, say at one round a client mis-clusters itself and updates its model incorrectly. At the next step, due to communication with all other clients, it can check the gradients of all other clients at its current model, have a chance to cluster correctly at this step, and update its model towards the optimum, regardless of the previous clustering error. Recall on the other hand that an algorithm like Myopic-Clustering (Alg. 1), while communication efficient ($N$ rounds per step), may not recover from past clustering mistakes since it doesn't check gradients rigorously in the same way. In the next section, we propose a more communication-efficient algorithm, Momentum-Clustering (Algorithm 4), which clusters momentums instead of gradients (reducing variance and thus clustering error) and requires only $\mathcal{O}(N)$ communication rounds per step.

---

[1]The batch-size constraint reduces variance of the stochastic gradients (Lemma 6). In Section 3.3 we propose another algorithm Momentum-Clustering for which there is no batch-size restriction and which reduces variance by clustering momentums instead of gradients.

[2]$g_i(x_{j,t-1}) = \frac{1}{|B_i|} \sum_b g_i(x_{j,t-1}; b)$, where $g_i(x_{j,t-1}; b)$ is the gradient computed using sample $b$ in the batch.

---

**Algorithm 2** Federated-Clustering

---

**Input** Learning rate: $\eta$. Initial parameters for each client: $\{x_{1,0}, ..., x_{N,0}\}$. Batch-size $|B_i|$ (see Theorem 3 for a lower bound on this quantity)[1]

1: **for** client $i \in [N]$ **do**
2:      Send $x_{i,0}$ to all clients $j \neq i$.
3: **for** round $t \in [T]$ **do**
4:      **for** client $i$ in [N] **do**
5:          Compute $g_i(x_{j,t-1})$ with batch-size $|B_i|$[2]and send to client $j$ for all $j \neq i \in [N]$.
6:          Compute $v_{i,t} \leftarrow$ `Threshold-Clustering`$(\{g_j(x_{i,t-1})\}_{j \in [N]}; 1$ cluster$; g_i(x_{i,t-1}))$.
7:          Update parameter: $x_{i,t} = x_{i,t-1} - \eta v_{i,t}$.
8:          Send $x_{i,t}$ to all clients $j \neq i$.
9: **Output:** Personalized parameters: $\{x_{1,T}, ..., x_{N,T}\}$.

---

**Algorithm 3** Threshold-Clustering

---

**Input** Points to be clustered: $\{z_1, ..., z_N\}$. Number of clusters: $K$. Cluster-center initializations: $\{v_{1,0}, ..., v_{K,0}\}$.

1: **for** round $l \in [M]$ **do**
2:      **for** cluster $k$ in [K] **do**
3:          Set radius $\tau_{k,l}$.
4:          Update cluster-center estimate:

$$v_{k,l} = \frac{1}{N} \sum_{i=1}^{N} \left( z_i \mathbb{1}(\|z_i - v_{k,l-1}\| \leq \tau_{k,l}) + v_{k,l-1} \mathbb{1}(\|z_i - v_{k,l-1}\| > \tau_{k,l}) \right). \quad (4)$$

5: **Output:** Cluster-center estimates $\{v_1 = v_{1,M}, ..., v_K = v_{K,M}\}$.

---

### 3.3 Improving Communication Overhead with Momentum

Federated-Clustering is inefficient, requiring $N^2$ rounds of communication between clients at each step (each client computes their gradient at every other client's parameter). Since momentums change much more slowly from round-to-round than gradients, a past clustering mistake will not have as much of a harmful impact on future correct clustering and convergence as when clustering gradients.

In Algorithm 4, at each step each client computes their momentum and sends it to the server. The server clusters the $N$ momentums, computes an update per-cluster, and sends the update to the clients in each cluster. Therefore, communication is limited to $N$ rounds per step.

#### 3.3.1 Analysis of Momentum-Clustering

The analysis of the momentum based method requires adapting the intra-cluster similarity and inter-cluster separation assumptions from before.

- **Assumption 8** (Intra-cluster Similarity): For all $i \sim j$ and $t \in [T]$,

$$m_{i,t} \overset{i.i.d.}{\sim} m_{j,t},$$

     where $m_{i,t}$ is defined as in (6).

- **Assumption 9** (Inter-cluster Separation): For all $i \not\sim j$ and $t \in [T]$,

$$\|\mathbb{E}m_{i,t} - \mathbb{E}m_{j,t}\|^2 \geq \Delta^2.$$

Note that the intra-cluster similarity assumption in this momentum setting is stronger than in the gradient setting (Assumption 4): namely we require that the momentum of clients in the same cluster be i.i.d. at all

---

**Algorithm 4** Momentum-Clustering

---

**Input** Learning rate: $\eta$. Initial parameters: $\{x_{1,0} = ... = x_{N,0}\}$.

1: **for** round $t \in [T]$ **do**
2:     **for** client $i$ in [N] **do**
3:         Client $i$ sends

$$m_{i,t} = \alpha g_i(x_{i,t-1}) + (1 - \alpha)m_{i,t-1} \tag{6}$$

    to server.
4:         Server generates cluster centers

$$\{v_{k,t}\}_{k\in[K]} \leftarrow \texttt{Threshold-Clustering}(\{m_{i,t}\}; K \text{ clusters}; \{v_{k,t-1}\}_{k\in[K]})$$

    and sends $v_{k_i,t}$ to client $i$, where $k_i$ denotes the cluster to which $i$ is assigned in this step.
5:         Client $i$ computes update: $x_{i,t} = x_{i,t-1} - \eta v_{k_i,t}$.
6: **Output:** Personalized parameters: $\{x_{1,T}, ..., x_{N,T}\}$.

---

points. This stronger assumption is the price we pay for a simpler and more practical algorithm. Finally, due to the fact that momentums are low-variance counterparts of gradients (Lemma 13), we can eliminate constraints on the batch size and still achieve the same rate.

**Theorem 4.** *Let* Assumptions [6-9] *hold with inter-cluster separation parameter* $\Delta \gtrsim \sigma/\delta_i$. *Under these conditions, suppose we run Algorithm 4 for $T$ rounds with learning rate* $\eta \lesssim \min\left\{\frac{1}{L}, \sqrt{\frac{\mathbb{E}(f_i(x_{i,0}) - f_i^*)}{LT(\sigma^2/n_i + \sigma^3/\Delta)}}\right\}$ *($f_i^*$ is the global minimum of $f_i$), fraction of bad clients* $\beta_i \lesssim \delta_i$, *and momentum parameter* $\alpha \gtrsim L\eta$. *If, in each round* $t \in [T]$, *we cluster with radius* $\tau \approx \sqrt{\delta_i\sigma\Delta}$ *for*

$$l \geq \max\left\{1, \max_{i\in N]} \frac{\log(\sqrt{\alpha}\sigma/\Delta)}{\log(1 - \delta_i/2)}\right\}$$

*steps, then for all* $i \in [N]$

$$\frac{1}{T}\sum_{t=1}^{T} \mathbb{E}\|\nabla f_i(x_{i,t-1})\|^2 \lesssim \sqrt{\frac{\sigma^2/n_i + \sigma^3/\Delta}{T}} + \frac{\beta_i\sigma\Delta}{T^{\frac{1}{4}}(\sigma^2/n_i + \sigma^3/\Delta)^{\frac{1}{4}}}. \tag{5}$$

*Proof.* See A.5. □

We see from (5) that when there are no malicious clients ($\beta_i = 0$), Momentum-Clustering achieves the same $\sqrt{\sigma^2/n_iT}$ convergence rate observed in (3), with no restrictions on the batch size.

## 4 Experiments

In this section, we first use a synthetic dataset to verify the assumptions and rates claimed in our theoretical analysis in the previous section; and second, we use the MNIST dataset (LeCun et al., 2010) and CIFAR dataset (Krizhevsky, 2009) to compare our proposed algorithm, Federated-Clustering, with existing state-of-the-art federated learning algorithms. All algorithms are implemented with PyTorch (Paszke et al., 2017).

### 4.1 Synthetic dataset

**Construction of synthetic dataset.** We consider a synthetic linear regression task with squared loss for which we construct $K = 4$ clusters, each with $n_i = 75$ clients. Clients in cluster $k \in [K]$ share the same minimizer $x_k^\star \in \mathbb{R}^d$. For each client $i$ in cluster $k$, we generate a sample matrix $A_i \in \mathbb{R}^{d\times n}$ from $\mathcal{N}(k, \mathbf{1}_{d\times n})$ and compute the associated target as $y_i = A_i^\top x_k^\star \in \mathbb{R}^n$. We choose the model dimension $d = 10$ to be greater than the number of local samples $n = 9$ such that the local linear system $y_i = A_i^\top x$ is overdetermined and the

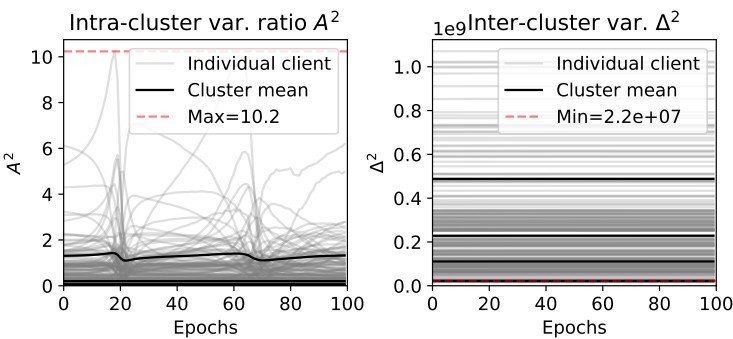

Figure 2: Here, we show empirically on a synthetic dataset that the intra-cluster variance ratio (7) is upper-bounded by a constant (left subplot) and the inter-cluster variance (8) that is lower-bounded by a constant (right subplot).

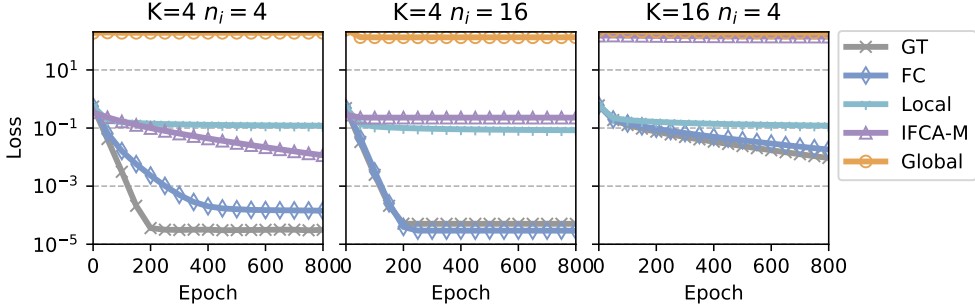

Figure 3: The performance of our algorithm vs. baselines on a synthetic dataset. When $n_i$ is small, the ground-truth outperforms our algorithm, but this difference vanishes with increasing $n_i$. This behavior is consistent with the dependence of the convergence rate on $n_i$ in Theorem 3: increasing $n_i$ improves convergence.

error $\|x^\star - x\|_2^2$ is large. A desired federated clustering algorithm determines the minimizer by incorporating information from other clients $j \sim i$ in the same cluster.

**Estimating constants in Assumptions 4 and 5.** In Fig. 2, using the above synthetic dataset

- we estimate the intra-cluster variance ratio $A^2$ by finding the upper bound of (7)

$$\frac{\|\nabla f_i(x) - \nabla \bar{f}_i(x)\|_2^2}{\|\nabla \bar{f}_i(x)\|_2^2};\tag{7}$$

- we estimate the inter-cluster variance $\Delta^2$ by setting $D = 0$ and computing the lower bound of (8)

$$\|\nabla f_i(x) - \nabla f_j(x)\|_2^2.\tag{8}$$

We run Federated-Clustering with perfect clustering assignments and estimate these bounds over time. The result is shown in Fig. 2 where grey lines are the quantities in (7) and (8) for individual clients, black lines are those quantities averaged within clusters, and red dashed lines are empirical bounds on the quantities. The left figure demonstrates that the intra-cluster variance ratio does not grow with time and can therefore reasonably be upper bounded by a constant $A^2$. Similarly, the right figure shows that the inter-cluster variance can be reasonably lower bounded by a positive constant $\Delta^2$. These figures empirically demonstrate that Assumptions 4 and 5 are realistic in practice.

Table 1: Comparison of test losses and accuracies for federated personalization algorithms on MNIST. FC outperforms all non-oracle baselines on two learning tasks.

|  | Rotation | | Private label | |
| --- | --- | --- | --- | --- |
|  | Acc.(%) | Loss | Acc.(%) | Loss |
| Local | 71.3 | 0.517 | 75.2 | 0.489 |
| Global | 46.6 | 0.631 | 22.2 | 0.803 |
| Ditto | 62.0 | 0.576 | 61.7 | 0.578 |
| IFCA | 54.6 | 0.588 | 65.4 | 0.531 |
| KNN | 52.1 | 2.395 | 63.2 | 1.411 |
| FC (ours) | **75.4** | **0.475** | **77.0** | **0.468** |
| GT (oracle) | 84.7 | 0.432 | 85.1 | 0.430 |

**Performance.** In Fig. 3, we compare the performance of our algorithm Federated-Clustering (FC) with several baselines: standalone training (Local), IFCA (Ghosh et al., 2020), FedAvg (Global) (McMahan et al., 2017), and distributed training with ground truth (GT) cluster information. We consider the synthetic dataset from before, starting with cluster parameters $(K, n_i) = (4, 4)$ and observe performance when increasing parameters to $n_i = 16$ and $K = 16$ separately. In each step of optimization, we run Threshold-Clustering for $l = 10$ rounds so that heuristically the outputs are close enough to cluster centers, cf. Fig. 7. We tune the learning rate separately for each algorithm through grid search, but preserve all other algorithmic setups. Our algorithm outperforms the non-oracle baselines in all cases. While Federated-Clustering is slightly worse than ground truth when $(K, n_i) = (4, 4)$, their performances are almost identical in the middle subplot for $n_i = 16$. This observation is consistent with the $n_i$-scaling observed in (3): as the number of clients-per-cluster increases, convergence improves.

## 4.2 MNIST experiment

In this section, we compare Federated-Clustering to existing federated learning baselines on the MNIST dataset. The dataset is constructed as follows, similar to Ghosh et al. (2020). The data samples are randomly shuffled and split into $K = 4$ clusters with $n_i = 75$ clients in each cluster. We consider two different tasks: 1) the *rotation* task transforms images in cluster $k$ by $k * 90$ degrees; 2) the *private label* task transforms labels in cluster $k$ with $T_k(y) : y \mapsto (y + k \mod 10)$, such that the same image may have different labels from cluster-to-cluster.

**Algorithm hyperparameters.** For these two experimental tasks, in addition to the baselines from our synthetic experiment, we include the KNN-personalization (Marfoq et al., 2021) and Ditto (Li et al., 2021) algorithms which both interpolate between a local and global model. The KNN-personalization is a linear combination of a global model, trained with FedAvg (McMahan et al., 2017), and a local model which is the aggregation of nearest-neighbor predictions in the client's local dataset to the global model's prediction. We set the coefficients of this linear combination to be $\lambda_{\text{knn}} = 0.5$ and $\lambda_{\text{knn}} = 0.9$ for the *rotation* and *private label* tasks, respectively. The Ditto objective is a personalized loss with an added regularization term that encourages closeness between the personalized and global models. Since tuning this regularization parameter $\lambda_{\text{ditto}}$ leads to a degenerated "Local" training where $\lambda_{\text{ditto}} = 0$, we fix $\lambda_{\text{ditto}} = 1$ for both *rotation* and *private label* tasks. To reduce the computation cost of our algorithm, in each iteration we randomly divide the $N$ clients into 16 subgroups and apply Federated-Clustering to each subgroup simultaneously. The clipping radius $\tau_{k,l}$ for each cluster $k$ is adaptively chosen to be the 20th-percentile of distances to the cluster-center.

**Performance.** The experimental results are listed in Table 1. Since an image can have different labels across clusters in the *private label* task, a model trained over the pool of all datasets only admits inferior performance. Therefore, distributed training algorithms that maintain a global model, such as FedAvg, Ditto, and KNN, perform poorly compared to training alone. On the other hand, our algorithm Federated-Clustering outperforms standalone training and all personalization baselines. This experiment suggests that our algorithm successfully explored the cluster structure and benefited from collaborative training.

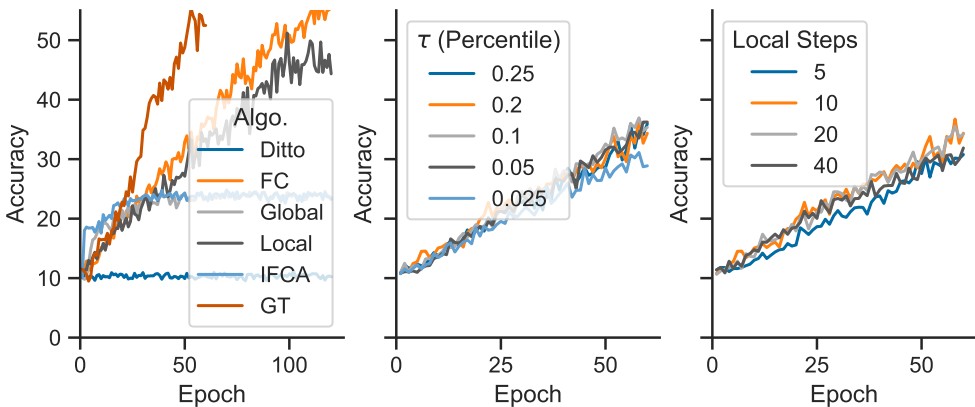

Figure 4: Performance of algorithms on CIFAR-10 dataset with the *private labels* task. **Left**: Relative accuracy of clustering algorithms. Algorithms optimized for global model performance, such as Ditto, IFCA, Global (FedAvg), perform poorly on personalization. FC outperforms Local training, showing that it benefits from collaboration between clients, and is competitive with GroundTruth. **Middle**: Impact of thresholding radius $\tau$ on accuracy. $\tau$ is the percentile of gradients distances from the cluster-center. **Right**: Impact of local gradient steps between two clustering calls. Early on in training, clusters are less identifiable so local optimization helps but these gains lessen later on when gradients from different clusters drift apart and clusters are better defined.

### 4.3 CIFAR experiment

In this section, we evaluate the efficacy of various clustering algorithms on the CIFAR-10 and CIFAR-100 datasets (Krizhevsky, 2009).

#### 4.3.1 CIFAR-10

For the CIFAR-10 experiment, we create 4 clusters, each containing 5 clients and transform the labels in each cluster such that different clusters can have different labels for the same image (the *private label* task in Section 4.2). We train a VGG-16 model (Simonyan & Zisserman, 2015) with batch size 32, learning rate 0.1, and momentum 0.9. The outcomes are presented in Fig. 4. The left subplot illustrates that collaborative clustering algorithms designed for global model training (e.g., Ditto, IFCA, Global) yield suboptimal models, as not all participating clients benefit from each other. On the other hand, Local and GroundTruth training are not influenced by the conflicting labels from other clusters so they significantly outperform Ditto and IFCA. Our Federated-Clustering (FC) algorithm also excludes such adversarial influence and, more importantly, outperforms Local training, showing that FC benefits from collaboration.

In the middle subplot, we examine the impact on accuracy of varying the thresholding radius $\tau$ (i.e. $\tau$ is set as the percentile of gradient distances from the cluster-center, so smaller percentile corresponds to smaller $\tau$). Our findings indicate that adopting a more conservative value for $\tau$ (lower percentile) does not substantially compromise accuracy.

The right subplot demonstrates the behavior of Federated-Clustering when the clustering oracle is invoked intermittently. The results suggest that increasing the number of local iterations boosts the learning curve early in training when gradients from different clusters are close together and clusters are ill-defined. However, this improvement plateaus when gradients become separated and clusters become well-defined.

#### 4.3.2 CIFAR-100

We consider the CIFAR-100 dataset distributed over 10 clusters so that each cluster contains 10 unique labels. In each cluster, we set 10 clients with IID data. We use a VGG-8 model for training and the same hyperparameters as those in the CIFAR-10 experiment (Section 4.3.1) and report the results in Fig. 5. While clients' data within each cluster share similar features, a small model like VGG-8 cannot sufficiently benefit

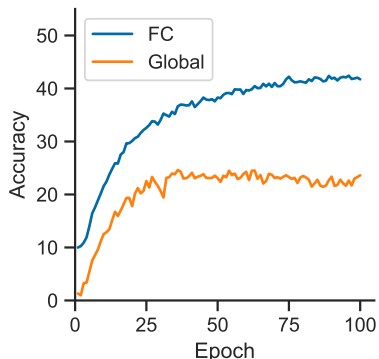

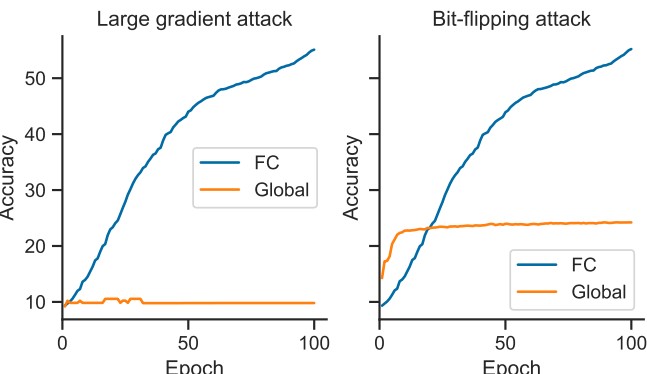

Figure 5: Performance of Federated-Clustering (FC) on the CIFAR-100 dataset. The Global (FedAvg) accuracy plateaus while FC continually improves.

Figure 6: Performance of Federated-Clustering (FC) against a large gradient attack (left) and bit-flipping attack (right). FC is robust to these attacks and significantly outperforms Global (FedAvg) performance.

from intra-cluster collaboration. Therefore the performance of Global training plateaus at a very low level while in contrast Federated-Clustering (FC) still benefits from collaboration and continues to improve over time.

### 4.4 Defense against Byzantine attacks

Byzantine attacks, in which attackers have full knowledge of the system and can deviate from the prescribed algorithm, are prevalent in distributed environments (Lamport et al., 2019). There are many forms of Byzantine attacks. For example, our *private label* setting in Sections 4.2 and 4.3.1 corresponds to the *label-flipping attack* in the Byzantine-robustness literature, since a malicious client can try to corrupt the model by assigning the wrong label to an image in training data.

In this section, we investigate two other attacks: Byzantine workers send either very large gradients or gradients with opposite signs. Using the MNIST dataset with the *private label* task, we set 4 clusters with 50 non-malicious clients each (so non-malicious clients from different clusters can have private labels) and add 50 Byzantine works to each cluster. We demonstrate the robustness of Federated-Clustering (FC) in Fig. 6. In both cases, Global training suffers from serious model degradation while FC successfully reaches high accuracy under these attacks, demonstrating its robustness.

### 4.5 Empirical Study on Clipping Iterations in Algorithm 3

We employ Algorithm 3 (Threshold-Clustering) on datasets to discern the effectiveness of the clipping iterations in identifying optimal cluster centers. These datasets share the same groundtruth cluster centers, and thus the same $\Delta$, but vary in their inter-cluster standard deviations, with $\sigma$ values of $0.5, 1, 2, 4$. Each dataset is made up of 90 ten-dimensional samples from 10 clusters, generated using the scikit-learn package (Pedregosa et al., 2011). For each iteration $l$ within cluster $k$, the clipping radius $\tau_{k,l}$ is defined as the 10th percentile of gradients' distances from the cluster-center. We repeat this experimental setup ten times for consistency.

The outcomes, presented in Fig. 7, show that the average distances initially decrease rapidly, then steadily approach convergence. To identify the *elbow* of a given curve $f$, we use the formula $\frac{f(l)-f(l-1)}{f(l-1)-\min_l f(l)}$, where curves post-*elbow* are notably flat. These *elbows* elucidate the correlation between $\sigma$ and $l$, indicating that for a fixed dataset (and its corresponding $\sigma$), one can pinpoint the minimal iterations $l$ needed for convergence. Notably, this observation appears to align with the $l \gtrsim \log \sigma$ lower bound stated in Theorem 1.

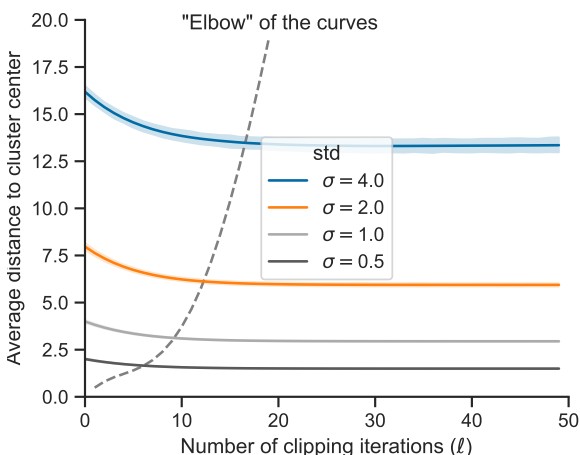

Figure 7: The average distance to cluster centers as a function of the number of clipping iterations $l$. The distance between cluster-centers ($\Delta$) is fixed while inter-cluster standard deviation $\sigma$ differs.

## 5 Conclusion

We develop gradient-based clustering algorithms to achieve personalization in federated learning. Our algorithms have optimal convergence guarantees. They asymptotically match the achievable rates when the true clustering of clients is known, and our analysis holds under light assumptions (e.g., for all smooth convex and non-convex losses). Furthermore, our algorithms are provably robust in the Byzantine setting where some fraction of the clients can arbitrarily corrupt their gradients. Future directions involve developing bespoke analysis for the convex-loss case and developing more communication-efficient versions of our algorithms. Further, our analysis can be used to show that our algorithms are incentive-compatible and lead to *stable coalitions* as in Donahue & Kleinberg (2021). This would form a strong argument towards encouraging participants in a federated learning system. Investigating such incentives and fairness concerns is another promising future direction.

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

# A  Proofs

## A.1  Proof of Theorem 1

First we establish some notation.

**Notation.**

- $\mathcal{G}_i$ are the good points and $\mathcal{B}_i$ the bad points from point $z_i$'s perspective. Therefore $|\mathcal{G}_i| + |\mathcal{B}_i| = N$.

- $k_i$ denotes the cluster to which client $i$ is assigned at the end of Threshold-Clustering.

- To facilitate the proof, we introduce a variable $c_{k_i,l}^2$ that quantifies the distance from the cluster-center-estimates to the true cluster means at each step of thresholding. Specifically, for client $i$'s cluster $k_i$ at round $l$ of Threshold-Clustering, we set

$$c_{k_i,l}^2 = \mathbb{E}\|v_{k_i,l-1} - \mathbb{E}z_i\|^2.$$

- For client $i$'s cluster $k_i$ at round $l$ of Threshold-Clustering, we use thresholding radius

$$\tau_{k_i,l}^2 \approx c_{k_i,l}^2 + \delta_i \sigma \Delta.$$

- We introduce a variable $y_{j,l}$ to denote the points clipped by Threshold-Clustering:

$$v_{k,l} = \frac{1}{N} \sum_{j \in [N]} \underbrace{z_j \mathbb{1}(\|z_j - v_{k,l-1}\| \le \tau_{k,l}) + v_{k,l-1} \mathbb{1}(\|z_j - v_{k,l-1}\| > \tau_{k,l})}_{y_{j,l}}.$$

*Proof of Theorem 1.* We prove the main result with the following sequence of inequalities, and then justify the labeled steps afterwards.

$$\mathbb{E}\|v_{k_i,l} - \mathbb{E}z_i\|^2 = \mathbb{E}\left\|\frac{1}{N}\sum_{j\in[N]} y_{j,l} - \mathbb{E}z_i\right\|^2$$

$$= \mathbb{E}\left\|(1-\beta_i)\left(\frac{1}{|\mathcal{G}_i|}\sum_{j\in\mathcal{G}_i} y_{j,l} - \mathbb{E}z_i\right) + \beta_i\left(\frac{1}{|\mathcal{B}_i|}\sum_{j\in\mathcal{B}_i} y_{j,l} - \mathbb{E}z_i\right)\right\|^2$$

$$\overset{(i)}{\leq} (1+\beta_i)(1-\beta_i)^2\mathbb{E}\left\|\left(\frac{1}{|\mathcal{G}_i|}\sum_{j\in\mathcal{G}_i} y_{j,l}\right) - \mathbb{E}z_i\right\|^2 + \left(1+\frac{1}{\beta_i}\right)\beta_i^2\mathbb{E}\left\|\left(\frac{1}{|\mathcal{B}_i|}\sum_{j\in\mathcal{B}_i} y_{j,l}\right) - \mathbb{E}x_i\right\|^2$$

$$\lesssim \underbrace{\mathbb{E}\left\|\left(\frac{1}{|\mathcal{G}_i|}\sum_{j\in\mathcal{G}_i} y_{j,l}\right) - \mathbb{E}z_i\right\|^2}_{\mathcal{E}_1} + \beta_i\underbrace{\mathbb{E}\left\|\left(\frac{1}{|\mathcal{B}_i|}\sum_{j\in\mathcal{B}_i} y_{j,l}\right) - \mathbb{E}z_i\right\|^2}_{\mathcal{E}_2}$$

$$\overset{(ii)}{\lesssim} \left((1-\delta_i)c_{k_i,l}^2 + \frac{\sigma^2}{n_i} + \frac{\sigma^3}{\Delta}\right) + \beta_i(c_{k_i,l}^2 + \delta_i\sigma\Delta)$$

$$\overset{(iii)}{\lesssim} (1-\delta_i/2)c_{k_i,l}^2 + \left(\frac{\sigma^2}{n_i} + \frac{\sigma^3}{\Delta} + \beta_i\sigma\Delta\right)$$

$$\overset{(iv)}{\lesssim} (1-\delta_i/2)^l c_{k_i,1}^2 + \left(\frac{\sigma^2}{n_i} + \frac{\sigma^3}{\Delta} + \beta_i\sigma\Delta\right)\sum_{q=0}^{l-1}(1-\delta_i/2)^q$$

$$\overset{(v)}{\lesssim} (1-\delta_i/2)^l\sigma^2 + \left(\frac{\sigma^2}{n_i} + \frac{\sigma^3}{\Delta} + \beta_i\sigma\Delta\right)$$

$$\overset{(vi)}{\lesssim} \frac{\sigma^2}{n_i} + \frac{\sigma^3}{\Delta} + \beta_i\sigma\Delta. \tag{9}$$

Justifications for the labeled steps are:

- (i) Young's inequality: $\|x+y\|^2 \leq (1+\epsilon)\|x\|^2 + (1+1/\epsilon)\|y\|^2$ for any $\epsilon > 0$.

- (ii) We prove this bound in Lemmas 1 and 5. Importantly, it shows that the clustering error is composed of two quantities: $\mathcal{E}_1$, the error contributed by good points from the cluster's perspective, and $\mathcal{E}_2$, the error contributed by the bad points from the cluster's perspective.

- (iii) Assumption that $\beta_i \lesssim \delta_i$

- (iv) Since $\mathbb{E}\|v_{k_i,l} - \mathbb{E}z_i\|^2 = c_{k_i,l+1}^2$, the inequality forms a recursion which we unroll over $l$ steps.

- (v) Assumption that $c_{k_i,1}^2 = \mathbb{E}\|v_{k_i,0} - \mathbb{E}z_i\|^2 \leq \sigma^2$. Also, the partial sum in the second term can be upper-bounded by a large-enough constant.

- (vi) Assumption that $l \geq \max\left\{1, \frac{\log(\sigma/\Delta)}{\log(1-\delta_i/2)}\right\}$

From (9), we see that $c_{k_i,l}^2 \lesssim \frac{\sigma^2}{n_i} + \frac{\sigma^3}{\Delta} + \beta_i\sigma\Delta$. Plugging this into the expression for $\tau_{k_i,l}^2$ gives $\tau_{k_i,l}^2 \approx \frac{\sigma^2}{n_i} + \frac{\sigma^3}{\Delta} + \delta_i\sigma\Delta \approx \delta_i\sigma\Delta$ for large $n_i$ and $\Delta$. $\qquad\square$

**Lemma 1** (Clustering Error due to Good Points).

$$\mathbb{E}\left\|\left(\frac{1}{|\mathcal{G}_i|}\sum_{j\in\mathcal{G}_i} y_{j,l}\right) - \mathbb{E}z_i\right\|^2 \lesssim (1-\delta_i)c_{k_i,l}^2 + \frac{\sigma^2}{n_i} + \frac{\sigma^3}{\Delta}$$

*Proof of Lemma 1.* We prove the main result with the following sequence of inequalities and justify the labeled steps afterward.

$$
\mathbb{E}\left\|\left(\frac{1}{|\mathcal{G}_i|}\sum_{j\in\mathcal{G}_i} y_{j,l}\right) - \mathbb{E}z_i\right\|^2 = \mathbb{E}\left\|\left(\frac{1}{|\mathcal{G}_i|}\sum_{j\in\mathcal{G}_i:j\sim i}(y_{j,l}-\mathbb{E}z_j)\right) + \left(\frac{1}{|\mathcal{G}_i|}\sum_{j\in\mathcal{G}_i:j\nsim i}(y_{j,l}-\mathbb{E}z_i)\right)\right\|^2
$$

$$
\overset{(i)}{\le} \left(1+\frac{2}{\delta_i}\right)\mathbb{E}\left\|\frac{1}{|\mathcal{G}_i|}\sum_{j\in\mathcal{G}_i:j\sim i}(y_{j,l}-\mathbb{E}z_j)\right\|^2 + \left(1+\frac{\delta_i}{2}\right)\mathbb{E}\left\|\frac{1}{|\mathcal{G}_i|}\sum_{j\in\mathcal{G}_i:j\nsim i}(y_{j,l}-\mathbb{E}z_i)\right\|^2
$$

$$
\overset{(ii)}{\lesssim} \left(1+\frac{2}{\delta_i}\right)\mathbb{E}\left\|\frac{1}{|\mathcal{G}_i|}\sum_{j\in\mathcal{G}_i:j\sim i}(\mathbb{E}y_{j,l}-\mathbb{E}z_j)\right\|^2 + \left(1+\frac{2}{\delta_i}\right)\mathbb{E}\left\|\frac{1}{|\mathcal{G}_i|}\sum_{j\in\mathcal{G}_i:j\sim i}(y_{j,l}-\mathbb{E}y_{j,l})\right\|^2
$$

$$
+ \left(1+\frac{\delta_i}{2}\right)\mathbb{E}\left\|\frac{1}{|\mathcal{G}_i|}\sum_{j\in\mathcal{G}_i:j\nsim i}(y_{j,l}-\mathbb{E}z_i)\right\|^2
$$

$$
\le \left(1+\frac{2}{\delta_i}\right)\delta_i^2\|\mathbb{E}_{j\in\mathcal{G}_i:j\sim i}(y_{j,l}-z_j)\|^2 + \left(1+\frac{2}{\delta_i}\right)\mathbb{E}\left\|\frac{1}{|\mathcal{G}_i|}\sum_{j\in\mathcal{G}_i:j\sim i}(y_{j,l}-\mathbb{E}y_{j,l})\right\|^2
$$

$$
+ \left(1+\frac{\delta_i}{2}\right)(1-\delta_i)^2\mathbb{E}_{j\in\mathcal{G}_i:j\nsim i}\|y_{j,l}-\mathbb{E}z_i\|^2
$$

$$
\lesssim \delta_i \underbrace{\|\mathbb{E}_{j\in\mathcal{G}_i:j\sim i}(y_{j,l}-z_j)\|^2}_{\mathcal{T}_1} + \left(1+\frac{2}{\delta_i}\right)\mathbb{E}\underbrace{\left\|\frac{1}{|\mathcal{G}_i|}\sum_{j\in\mathcal{G}_i:j\sim i}(y_{j,l}-\mathbb{E}y_{j,l})\right\|^2}_{\mathcal{T}_2}
$$

$$
+ \left(1+\frac{\delta_i}{2}\right)(1-\delta_i)^2\underbrace{\mathbb{E}_{j\in\mathcal{G}_i:j\nsim i}\|y_{j,l}-\mathbb{E}z_i\|^2}_{\mathcal{T}_3}
$$

$$
\overset{(iii)}{\lesssim} \delta_i\left(c_{k_i,l}^2 + \frac{(c_{k_i,l}^2+\sigma^2)\sigma}{\delta_i\Delta}\right) + \left(1+\frac{2}{\delta_i}\right)\frac{n_i}{|\mathcal{G}_i|^2}\sigma^2
$$

$$
+ (1-\delta_i)^2\left(1+\frac{\delta_i}{2}\right)\left(\left(1+\frac{\delta_i}{2}+\frac{\sigma^2}{\delta_i\Delta^2}\right)c_{k_i,l}^2 + \frac{\sigma^3}{\Delta}\right)
$$

$$
\lesssim \left(1-\delta_i+\frac{\sigma}{\Delta}+\frac{\sigma^2}{\delta_i\Delta^2}\right)c_{k_i,l}^2 + \left(\frac{\sigma^2}{n_i}+\frac{\sigma^3}{\Delta}\right)
$$

$$
\overset{(iv)}{\lesssim} (1-\delta_i)c_{k_i,l}^2 + \left(\frac{\sigma^2}{n_i}+\frac{\sigma^3}{\Delta}\right).
$$

- (i), (ii) Young's inequality

- (iii) We prove this bound in Lemmas 2, 3, and 4. Importantly, it shows that, from point $i$'s perspective, the error of its cluster-center-estimate is composed of three quantities: $\mathcal{T}_1$, the error introduced by our thresholding procedure on the good points which belong to $i$'s cluster (and therefore ideally are included within the thresholding radius); $\mathcal{T}_2$, which accounts for the variance of the points in $i$'s cluster; and $\mathcal{T}_3$, the error due to the good points which don't belong to $i$'s cluster (and therefore ideally are forced outside the thresholding radius).

- (iv) Assumption that $\Delta \gtrsim \sigma/\delta_i$.

$\square$

**Lemma 2** (Bound $\mathcal{T}_1$: Error due to In-Cluster Good Points)**.**

$$
\|\mathbb{E}_{j\in\mathcal{G}_i:j\sim i}(y_{j,l}-z_j)\|^2 \lesssim c_{k_i,l}^2 + \frac{(c_{k_i,l}^2+\sigma^2)\sigma}{\delta_i\Delta}.
$$

*Proof of Lemma 2.* By definition of $y_{j,l}$,

$$
\begin{aligned}
\mathbb{E}_{j \in \mathcal{G}_i : j \sim i} \|y_{j,l} - z_j\| &= \mathbb{E}[\|v_{k_i,l-1} - z_j\| \mathbb{1}(\|v_{k_i,l-1} - z_j\| > \tau_{k_i,l})] \\
&\leq \frac{\mathbb{E}[\|v_{k_i,l-1} - z_j\|^2 \mathbb{1}(\|v_{k_i,l-1} - z_j\| > \tau_{k_i,l})]}{\tau_{k_i,l}} \\
&\leq \frac{\mathbb{E}\|v_{k_i,l-1} - z_j\|^2}{\tau_{k_i,l}} \\
&\lesssim \frac{\mathbb{E}\|v_{k_i,l-1} - \mathbb{E}z_i\|^2 + \mathbb{E}\|\mathbb{E}z_j - z_j\|^2}{\tau_{k_i,l}} \\
&\leq \frac{c_{k_i,l}^2 + \sigma^2}{\tau_{k_i,l}}
\end{aligned}
$$

Finally, by Jensen's inequality and plugging in the value for $\tau_{k_i,l}$,

$$
\begin{aligned}
\|\mathbb{E}(y_{j,l} - z_j)\|^2 &\leq (\mathbb{E}\|y_{j,l} - z_j\|)^2 \\
&\lesssim \frac{(c_{k_i,l}^2 + \sigma^2)^2}{\tau_{k_i,l}^2} \\
&\lesssim \frac{c_{k_i,l}^4}{c_{k_i,l}^2} + \frac{c_{k_i,l}^2 \sigma^2}{\delta_i \sigma \Delta} + \frac{\sigma^4}{\delta_i \sigma \Delta} \\
&= c_{k_i,l}^2 + \frac{(c_{k_i,l}^2 + \sigma^2)\sigma}{\delta_i \Delta}.
\end{aligned}
$$

$\square$

**Lemma 3** (Bound $\mathcal{T}_2$: Variance of Clipped Points).

$$
\mathbb{E}\left\| \frac{1}{|\mathcal{G}_i|} \sum_{j \in \mathcal{G}_i : j \sim i} (y_{j,l} - \mathbb{E}y_{j,l}) \right\|^2 \leq \frac{n_i}{|\mathcal{G}_i|^2} \sigma^2.
$$

*Proof of Lemma 3.* Note that the elements in the sum $\sum_{j \in \mathcal{G}_i : j \sim i}(y_{j,l} - \mathbb{E}y_{j,l})$ are not independent. Therefore, we cannot get rid of the cross terms when expanding the squared-norm. However, if for each round of thresholding we were sample a fresh batch of points to set the new cluster-center estimate, then the terms would be independent. With this resampling strategy, our bounds would only change by a constant factor. Therefore, for ease of analysis, we will assume the terms in the sum are independent. In that case,

$$
\begin{aligned}
\mathbb{E}\left\| \frac{1}{|\mathcal{G}_i|} \sum_{j \in \mathcal{G}_i : j \sim i} (y_{j,l} - \mathbb{E}y_{j,l}) \right\|^2 &\leq \frac{n_i}{|\mathcal{G}_i|^2} \mathbb{E}\|y_{j,l} - \mathbb{E}y_{j,l}\|^2 \\
&\leq \frac{n_i}{|\mathcal{G}_i|^2} \mathbb{E}\|z_j - \mathbb{E}z_j\|^2 \\
&\leq \frac{n_i}{|\mathcal{G}_i|^2} \sigma^2,
\end{aligned}
$$

where the second-to-last inequality follows from the contractivity of the thresholding procedure. $\square$

**Lemma 4** (Bound $\mathcal{T}_3$: Error due to Out-of-Cluster Good Points).

$$
\mathbb{E}_{j \in \mathcal{G}_i : j \not\sim i} \|y_{j,l} - \mathbb{E}z_i\|^2 \lesssim \left( 1 + \frac{\delta_i}{2} + \frac{\sigma^2}{\delta_i \Delta^2} \right) c_{k_i,l}^2 + \frac{\sigma^3}{\Delta}.
$$

*Proof of Lemma 4.* By Young's inequality,

$$\mathbb{E}_{j \in \mathcal{G}_i : j \not\sim i} \|y_{j,l} - \mathbb{E}z_i\|^2 \leq \left(1 + \frac{\delta_i}{2}\right) \mathbb{E}\|v_{k_i,l-1} - \mathbb{E}z_i\|^2 + \left(1 + \frac{2}{\delta_i}\right) \mathbb{E}_{j \in \mathcal{G}_i : j \not\sim i} \|y_{j,l} - v_{k_i,l-1}\|^2$$

$$\leq \left(1 + \frac{\delta_i}{2}\right) c_{k_i,l}^2 + \left(1 + \frac{2}{\delta_i}\right) \mathbb{E}_{j \in \mathcal{G}_i : j \not\sim i} \|y_{j,l} - v_{k_i,l-1}\|^2$$

$$= \left(1 + \frac{\delta_i}{2}\right) c_{k_i,l}^2 + \left(1 + \frac{2}{\delta_i}\right) \mathbb{E}_{j \in \mathcal{G}_i : j \not\sim i} [\|z_j - v_{k_i,l-1}\|^2 \mathbb{1}\{\|z_j - v_{k_i,l-1}\| \leq \tau_{k_i,l}\}]$$

$$\leq \left(1 + \frac{\delta_i}{2}\right) c_{k_i,l}^2 + \left(1 + \frac{2}{\delta_i}\right) \tau_{k_i,l}^2 \mathbb{P}_{j \in \mathcal{G}_i : j \not\sim i}(\|z_j - v_{k_i,l-1}\| \leq \tau_{k_i,l}).$$

We now have to bound the probability in the expression above. Note that if $\|v_{k_i,l-1} - z_j\| \leq \tau_{k_i,l}$, then

$$\|\mathbb{E}z_j - \mathbb{E}z_i\|^2 \lesssim \|z_j - \mathbb{E}z_j\|^2 + \|z_j - \mathbb{E}v_{k_i,l-1}\|^2 + \|\mathbb{E}v_{k_i,l-1} - \mathbb{E}z_i\|^2$$

$$\lesssim \|z_j - \mathbb{E}z_j\|^2 + \|z_j - \mathbb{E}z_j\|^2 + \|\mathbb{E}z_j - \mathbb{E}v_{k_i,l-1}\|^2 + \mathbb{E}\|v_{k_i,l-1} - \mathbb{E}z_i\|^2 + \mathbb{E}\|z_i - \mathbb{E}z_i\|^2$$

$$\lesssim \|z_j - \mathbb{E}z_j\|^2 + \tau_{k_i,l}^2 + c_{k_i,l}^2 + \sigma^2.$$

By Assumption 2, this implies that

$$\Delta^2 \lesssim \|z_j - \mathbb{E}z_j\|^2 + \tau_{k_i,l}^2 + c_{k_i,l}^2 + \sigma^2$$

which means that

$$\|z_j - \mathbb{E}z_j\|^2 \gtrsim \Delta^2 - (\tau_{k_i,l}^2 + c_{k_i,l}^2 + \sigma^2).$$

By Markov's inequality,

$$\mathbb{P}(\|z_j - \mathbb{E}z_j\|^2 \gtrsim \Delta^2 - (\tau_{k_i,l}^2 + c_{k_i,l}^2 + \sigma^2)) \leq \frac{\sigma^2}{\Delta^2 - (\tau_{k_i,l}^2 + c_{k_i,l}^2 + \sigma^2)} \lesssim \frac{\sigma^2}{\Delta^2}$$

as long as

$$\Delta^2 \gtrsim \tau_{k_i,l}^2 + c_{k_i,l}^2 + \sigma^2,$$

which holds due to the constraint on $\Delta$ in the theorem statement. Therefore

$$\mathbb{E}_{j \in \mathcal{G}_i : j \not\sim i} \|y_{j,l} - \mathbb{E}z_i\|^2 \lesssim \left(1 + \frac{\delta_i}{2}\right) c_{k_i,l}^2 + \left(1 + \frac{2}{\delta_i}\right) \frac{(c_{k_i,l}^2 + \delta_i \sigma \Delta)\sigma^2}{\Delta^2}$$

$$\leq \left(1 + \frac{\delta_i}{2} + \frac{\sigma^2}{\delta_i \Delta^2}\right) c_{k_i,l}^2 + \frac{\sigma^3}{\Delta}.$$

$\square$

**Lemma 5** (Clustering Error due to Bad Points)**.**

$$\mathbb{E} \left\| \left(\frac{1}{|\mathcal{B}_i|} \sum_{j \in \mathcal{B}_i} y_{j,l}\right) - \mathbb{E}z_i \right\|^2 \lesssim c_{k_i,l}^2 + \delta_i \sigma \Delta$$

*Proof of Lemma 5.*

$$\mathbb{E} \left\| \left(\frac{1}{|\mathcal{B}_i|} \sum_{j \in \mathcal{B}_i} y_{j,l}\right) - \mathbb{E}z_i \right\|^2 \leq \mathbb{E}_{j \in \mathcal{B}_i} \|y_{j,l} - \mathbb{E}z_i\|^2$$

$$\lesssim \mathbb{E}_{j \in \mathcal{B}_i} \|y_{j,l} - v_{k_i,l-1}\|^2 + \mathbb{E}\|v_{k_i,l-1} - \mathbb{E}z_i\|^2$$

$$\lesssim c_{k_i,l}^2 + \delta_i \sigma \Delta.$$

The last inequality follows from the intuition that bad points will position themselves at the edge of the thresholding ball, a distance $\tau_{k_i,l}$ away from the current center-estimate $v_{k_i,l-1}$. Therefore we cannot do better than upper-bounding $\mathbb{E}_{j \in \mathcal{B}_i} \|y_{j,l} - v_{k_i,l-1}\|^2$ by $\tau_{k_i,l}^2 \approx c_{k_i,l}^2 + \delta_i \sigma \Delta$, the squared-radius of the ball. $\square$

### A.2 Proof of Theorem 2

*Proof of Theorem 2.* Let

$$\mathcal{D}_1 = \begin{cases} \delta & \text{w.p. } p \\ 0 & \text{w.p. } 1-p \end{cases}$$

and

$$\mathcal{D}_2 = \begin{cases} \delta & \text{w.p. } 1-p \\ 0 & \text{w.p. } p \end{cases}$$

and define the mixture $\mathcal{M} = \frac{1}{2}\mathcal{D}_1 + \frac{1}{2}\mathcal{D}_2$. Also consider the mixture $\tilde{\mathcal{M}} = \frac{1}{2}\tilde{\mathcal{D}}_1 + \frac{1}{2}\tilde{\mathcal{D}}_2$, where $\tilde{\mathcal{D}}_1 = 0$ and $\tilde{\mathcal{D}}_2 = \delta$. It is impossible to distinguish whether a sample comes from $\mathcal{M}$ or $\tilde{\mathcal{M}}$. Therefore, if you at least know a sample came from either $\mathcal{M}$ or $\tilde{\mathcal{M}}$ but not which one, the best you can do is to estimate $\mu_1$ with $\hat{\mu}_1 = \frac{\delta p}{2}$, half-way between the mean of $\mathcal{D}_1$, which is $\delta p$, and the mean of $\tilde{\mathcal{D}}_1$, which is 0. In this case

$$\mathbb{E}\|\hat{\mu}_1 - \mu_1\|^2 = \frac{\delta^2 p^2}{4}.$$

If $p \leq \frac{1}{2}$, then

$$\Delta = (1-p)\delta - p\delta = (1-2p)\delta. \tag{10}$$

Also,

$$\sigma^2 = \delta^2 p(1-p). \tag{11}$$

Equating $\delta^2$ in (10) and (11),

$$\frac{\Delta^2}{(1-2p)^2} = \frac{\sigma^2}{p(1-p)},$$

which can be rearranged to

$$(4\sigma^2 + \Delta^2)p^2 - (4\sigma^2 + \Delta^2)p + \sigma^2 = 0.$$

Solving for $p$,

$$p = \frac{1}{2} - \frac{\Delta}{2\sqrt{4\sigma^2 + \Delta^2}}. \tag{12}$$

Note that,

$$\frac{\delta^2 p^2}{4} = \frac{\sigma^2 p^2}{4p(1-p)}$$

$$= \frac{\sigma^2 p}{4(1-p)}. \tag{13}$$

Plugging the expression for $p$ from (12) into (13), we can see that

$$\frac{\delta^2 p^2}{4} = \frac{\sigma^2}{4}\left(\frac{\sqrt{4\sigma^2 + \Delta^2} - \Delta}{\Delta}\right) = \frac{\sigma^2}{4}\left(\sqrt{1 + \frac{4\sigma^2}{\Delta^2}} - 1\right) \geq \frac{\sigma^2}{4}\left(\frac{2\sigma^2}{\Delta^2} - \frac{2\sigma^4}{\Delta^4}\right).$$

The last step used an immediately verifiable inequality that $\sqrt{1+x} \geq 1 + \frac{x}{2} - \frac{x^2}{8}$ for all $x \in [0, 8]$. Finally, we can choose $\Delta^2 \geq 2\sigma^2$ to give the result that

$$\mathbb{E}\|\hat{\mu}_1 - \mu_1\|^2 \geq \frac{\delta^2 p^2}{4} \geq \frac{\sigma^4}{4\Delta^2}.$$

Finally, suppose that there is only a single cluster with $K = 1$. Then, given $n$ stochastic samples. standard information theoretic lower bounds show that we will have an error at least

$$\mathbb{E}\|\hat{\mu}_1 - \mu_1\|^2 \geq \frac{\sigma^2}{4n}.$$

Combining these two lower bounds yields the proof of the theorem. $\qquad\square$

### A.3 Proof of Theorem 3

First we establish some notation.

**Notation.**

- $\mathcal{G}_i$ are the good clients and $\mathcal{B}_i$ the bad clients from client $i$'s perspective. Therefore $|\mathcal{G}_i| + |\mathcal{B}_i| = N$.

- $\mathbb{E}_x$ denotes conditional expectation given the parameter, e.g. $\mathbb{E}_x g(x) = \mathbb{E}[g(x)|x]$. $\mathbb{E}$ denotes expectation over all randomness.

- $\overline{X_t} \triangleq \frac{1}{T} \sum_{t=1}^{T} X_t$ for a general variable $X_t$ indexed by $t$.

- We use $f_i(x)$ to denote average loss on a general batch $B$ of samples. That is, if $f_i(x;b)$ is the loss on a single sample $b$, we define $f_i(x) = \frac{1}{|B|} \sum_{b \in B} f_i(x;b)$.

- $\bar{f}_i(x) \triangleq \frac{1}{n_i} \sum_{j \in \mathcal{G}_i : j \sim i} f_j(x)$

- We introduce a variable $\rho^2$ to bound the variance of the gradients

$$\mathbb{E}\|g_i(x) - \mathbb{E}_x g_i(x)\|^2 \leq \rho^2,$$

  and show in Lemma 6 how this can be written in terms of the variance of the gradients computed over a batch size of 1.

- $l_t$ is the number of rounds that Threshold-Clustering is run in round $t$ of Federated-Clustering.

- $k_i$ denotes the cluster to which client $i$ is assigned.

- $v_{i,l,t}$ denotes the gradient update for client $i$ in round $t$ of Federated-Clustering and round $l$ of Threshold-Clustering. That is, $v_{i,l_t,t}$ corresponds to the quantity returned in Step 6 of Algorithm 2.

- To facilitate the proof, we introduce a variable $c_{k_i,l,t}$ that quantifies the distance from the cluster-center-estimates to the true cluster means. Specifically, for client $i$'s cluster $k_i$ at round $t$ of Federated-Clustering and round $l$ of Threshold-Clustering we set

$$c_{k_i,l,t}^2 = \mathbb{E}\|v_{i,l-1,t} - \mathbb{E}_x \bar{g}_i(x_{i,t-1})\|^2.$$

- For client $i$'s cluster $k_i$ at round $t$ of Federated-Clustering and round $l$ of Threshold-Clustering, we use thresholding radius

$$\tau_{k_i,l,t}^2 \approx c_{k_i,l,t}^2 + A^4 \mathbb{E}\|\nabla \bar{f}_i(x_{i,t-1})\|^2 + \delta_i \rho \Delta.$$

- Finally, we introduce a variable $y_{j,l,t}$ to denote the points clipped by Threshold-Clustering:

$$v_{i,l,t} = \frac{1}{N} \sum_{j \in [N]} \underbrace{\mathbb{1}(\|g_j(x_{i,t-1}) - v_{i,l-1,t}\| \leq \tau_{k_i^t,l}) + v_{i,l-1,t}\mathbb{1}(\|g_j(x_{i,t-1}) - v_{i,l-1,t}\| > \tau_{k_i^t,l})}_{y_{j,l,t}}.$$

*Proof of Theorem 3.* In this proof, our goal is to bound $\overline{\mathbb{E}\|\nabla f_i(x_{i,t-1})\|^2}$ for each client $i$, thus showing convergence. Recall that our thresholding procedure clusters the gradients of clients at each round and estimates the center of each cluster. These estimates are then used to update the parameters of the clusters. Therefore, we expect $\overline{\mathbb{E}\|\nabla f_i(x_{i,t-1})\|^2}$ to be bounded in terms of the error of this estimation procedure. The following sequence of inequalities shows this.

By $L$-smoothness of $f_i$ and setting $\eta \leq 1/L$,

$$f_i(x_{i,t}) \leq f_i(x_{i,t-1}) + \langle \nabla f_i(x_{i,t-1}), x_{i,t} - x_{i,t-1} \rangle + \frac{L}{2}\|x_{i,t} - x_{i,t-1}\|^2$$

$$= f_i(x_{i,t-1}) - \eta \langle \nabla f_i(x_{i,t-1}), v_{i,l_t,t} \rangle + \frac{L\eta^2}{2}\|v_{i,l_t,t}\|^2$$

$$= f_i(x_{i,t-1}) + \frac{\eta}{2}\|v_{i,l_t,t} - \nabla f_i(x_{i,t-1})\|^2 - \frac{\eta}{2}\|\nabla f_i(x_{i,t-1})\|^2 - \frac{\eta}{2}(1 - L\eta)\|v_{i,l_t,t}\|^2$$

$$\leq f_i(x_{i,t-1}) + \eta\|v_{i,l_t,t} - \nabla \bar{f}_i(x_{i,t-1})\|^2 + \eta A^2\|\nabla \bar{f}_i(x_{i,t-1})\|^2 - \frac{\eta}{2}\|\nabla f_i(x_{i,t-1})\|^2 - \frac{\eta}{2}(1 - L\eta)\|v_{i,l_t,t}\|^2. \tag{14}$$

Recall that $v_{i,l_t,t}$ is client $i$'s cluster-center estimate at round $t$ of optimization, so the second term on the right side of (14) is the error due to the clustering procedure. In Lemma 7, we show that, in expectation, this error is bounded by

$$\delta_i \overline{\mathbb{E}\|\nabla \bar{f}_i(x_{i,t-1})\|^2} + \frac{\rho}{\Delta}D^2\overline{\mathbb{E}\|\nabla f_i(x_{i,t-1})\|^2} + \left(\frac{\rho^2}{n_i} + \frac{\rho^3}{\Delta} + \beta_i \rho \Delta\right).$$

Therefore, subtracting $f_i^*$ from both sides, summing (14) over $t$, dividing by $T$, taking expectations, applying Lemma 7 to (14), and applying the constraint on $\Delta$ from the theorem statement, we have

$$\eta\overline{\mathbb{E}\|\nabla f_i(x_{i,t-1})\|^2} \lesssim \frac{\mathbb{E}(f_i(x_{i,0}) - f_i^*)}{T} + \eta A^2\overline{\mathbb{E}\|\nabla \bar{f}_i(x_{i,t-1})\|^2} + \eta\left(\frac{\rho^2}{n_i} + \frac{\rho^3}{\Delta} + \beta_i \rho \Delta\right). \tag{15}$$

The third term on the right side of (14) reflects the fact that clients in the same cluster may have different loss objectives. Far from their optima, these loss objectives may look very different and therefore be hard to cluster together.

In order to bound this term, we use a similar argument as above. By $L$-smoothness of $f_i$'s and setting $\eta \leq 1/L$,

$$\bar{f}_i(x_{i,t}) \leq \bar{f}_i(x_{i,t-1}) + \langle \nabla \bar{f}_i(x_{i,t-1}), x_{i,t} - x_{i,t-1} \rangle + \frac{L}{2}\|x_{i,t} - x_{i,t-1}\|^2$$

$$= \bar{f}_i(x_{i,t-1}) - \eta \langle \nabla \bar{f}_i(x_{i,t-1}), v_{i,l_t,t} \rangle + \frac{L\eta^2}{2}\|v_{i,l_t,t}\|^2$$

$$= \bar{f}_i(x_{i,t-1}) + \frac{\eta}{2}\|v_{i,l_t,t} - \nabla \bar{f}_i(x_{i,t-1})\|^2 - \frac{\eta}{2}\|\nabla \bar{f}_i(x_{i,t-1})\|^2 - \frac{\eta}{2}(1 - L\eta)\|v_{i,l_t,t}\|^2.$$

Subtracting $\bar{f}_i^*$ from both sides, summing over $t$, dividing by $T$, taking expectations, and applying Lemma 7,

$$\eta\overline{\mathbb{E}\|\nabla \bar{f}_i(x_{i,t-1})\|^2} \lesssim \frac{\mathbb{E}(\bar{f}_i(x_{i,0}) - \bar{f}_i^*)}{T} + \frac{\eta\rho}{\Delta}D^2\overline{\mathbb{E}\|\nabla f_i(x_{i,t-1})\|^2} + \eta\left(\frac{\rho^2}{n_i} + \frac{\rho^3}{\Delta} + \beta_i \rho \Delta\right). \tag{16}$$

Combining (15) and (16), and applying the constraint on $\Delta$ from the theorem statement, we have that

$$\eta\overline{\mathbb{E}\|\nabla f_i(x_{i,t-1})\|^2} \lesssim \frac{\mathbb{E}(f_i(x_{i,0}) - f_i^*) + A^2\mathbb{E}(\bar{f}_i(x_0) - \bar{f}_i^*)}{T} + \eta\max(1, A^2)\left(\frac{\rho^2}{n_i} + \frac{\rho^3}{\Delta} + \beta_i \rho \Delta\right). \tag{17}$$

Dividing both sides of (17) by $\eta = 1/L$ and noting that $\rho^2 = \sigma^2/|B|$ from Lemma 6,

$$\overline{\mathbb{E}\|\nabla f_i(x_{i,t-1})\|^2} \lesssim \frac{\mathbb{E}(f_i(x_{i,0}) - f_i^*) + A^2\mathbb{E}(\bar{f}_i(x_{i,0}) - \bar{f}_i^*)}{\eta T} + \max(1, A^2)\left(\frac{\rho^2}{n_i} + \frac{\rho^3}{\Delta} + \beta_i \rho \Delta\right) \tag{18}$$

$$\leq \frac{L(\mathbb{E}(f_i(x_{i,0}) - f_i^*) + A^2\mathbb{E}(\bar{f}_i(x_{i,0}) - \bar{f}_i^*))}{T} + \frac{\max(1, A^2)(\sigma^2/n_i + \sigma^3/\Delta + \beta_i \sigma \Delta)}{\sqrt{|B|}}$$

$$\lesssim \sqrt{\frac{\max(1, A^2)(\sigma^2/n_i + \sigma^3/\Delta + \beta_i \sigma \Delta)}{T}},$$

where the last inequality follows from setting

$$|B| \gtrsim \max(1, A^2)\left(\frac{\sigma^2}{n_i} + \frac{\sigma^3}{\Delta} + \beta_i \sigma \Delta\right)T.$$

Since $T = m_i/|B|$, this is equivalent to setting

$$|B| \gtrsim \sqrt{\max(1, A^2)\left(\frac{\sigma^2}{n_i} + \frac{\sigma^3}{\Delta} + \beta_i \sigma \Delta\right)m_i}.$$

$\square$

**Lemma 6** (Variance reduction using batches). *If, for a single sample $b$,*

$$\mathbb{E}_x \|g_i(x; b) - \mathbb{E}_x g_i(x; b)\|^2 \leq \sigma^2,$$

*then for a batch $B$ of samples,*

$$\mathbb{E}_x \|g_i(x) - \mathbb{E}_x g_i(x)\|^2 \leq \frac{\sigma^2}{|B|}.$$

*Proof of Lemma 6.* Due to the independence and unbiasedness of stochastic gradients,

$$\mathbb{E}_x \|g_i(x) - \mathbb{E}_x g_i(x)\|^2 = \frac{1}{|B|^2} \sum_{b \in B} \mathbb{E}_x \|g_i(x; b) - \mathbb{E}_x g_i(x; b)\|^2$$
$$\leq \frac{\sigma^2}{|B|}.$$

$\square$

**Lemma 7** (Bound on Clustering Error).

$$\overline{\mathbb{E}\|v_{i,l_t,t} - \nabla \bar{f}_i(x_{i,t-1})\|^2} \lesssim \delta_i \overline{\mathbb{E}\|\nabla \bar{f}_i(x_{i,t-1})\|^2} + \frac{\rho}{\Delta} D^2 \overline{\mathbb{E}\|\nabla f_i(x_{i,t-1})\|^2} + \left(\frac{\rho^2}{n_i} + \frac{\rho^3}{\Delta} + \beta_i \rho \Delta\right)$$

*Proof of Lemma 7.* We prove the main result with the following sequence of inequalities and justify the labeled steps afterwards.

$$
\begin{aligned}
\overline{\mathbb{E}\|v_{i,l_t,t} - \nabla \bar{f}_i(x_{i,t-1})\|^2} &= \overline{\mathbb{E}\|v_{i,l_t,t} - \mathbb{E}_x \bar{g}_i(x_{i,t-1})\|^2} \\
&= \overline{\mathbb{E}\left\| \frac{1}{N} \sum_{j \in [N]} y_{j,l_t,t} - \mathbb{E}_x \bar{g}_i(x_{i,t-1}) \right\|^2} \\
&= \overline{\mathbb{E}\left\| (1-\beta_i)\left( \frac{1}{|\mathcal{G}_i|} \sum_{j \in \mathcal{G}_i} y_{j,l_t,t} - \mathbb{E}_x \bar{g}_i(x_{i,t-1}) \right) + \beta_i \left( \frac{1}{|\mathcal{B}_i|} \sum_{j \in \mathcal{B}_i} y_{j,t,l} - \mathbb{E}_x \bar{g}_i(x_{i,t-1}) \right) \right\|^2} \\
&\overset{(i)}{\leq} (1+\beta_i)(1-\beta_i)^2 \overline{\mathbb{E}\left\| \left( \frac{1}{|\mathcal{G}_i|} \sum_{j \in \mathcal{G}_i} y_{j,l_t,t} \right) - \mathbb{E}_x \bar{g}_i(x_{i,t-1}) \right\|^2} \\
&\quad + \left(1 + \frac{1}{\beta_i}\right) \beta_i^2 \overline{\mathbb{E}\left\| \left( \frac{1}{|\mathcal{B}_i|} \sum_{j \in \mathcal{B}_i} y_{j,l_t,t} \right) - \mathbb{E}_x \bar{g}_i(x_{i,t-1}) \right\|^2} \\
&\leq \underbrace{\overline{\mathbb{E}\left\| \left( \frac{1}{|\mathcal{G}_i|} \sum_{j \in \mathcal{G}_i} y_{j,l_t,t} \right) - \mathbb{E}_x \bar{g}_i(x_{i,t-1}) \right\|^2}}_{\mathcal{E}_1} + \beta_i \underbrace{\overline{\mathbb{E}\left\| \left( \frac{1}{|\mathcal{B}_i|} \sum_{j \in \mathcal{B}_i} y_{j,l_t,t} \right) - \mathbb{E}_x \bar{g}_i(x_{i,t-1}) \right\|^2}}_{\mathcal{E}_2} \\
&\overset{(ii)}{\lesssim} (1-\delta_i+\beta_i)\overline{c_{k_i,l_t,t}^2} + (\delta_i + \beta_i A^4)\overline{\mathbb{E}\|\nabla \bar{f}_i(x_{i,t-1})\|^2} + \frac{\rho}{\Delta}D^2 \overline{\mathbb{E}\|\nabla f_i(x_{i,t-1})\|^2} \\
&\quad + \left( \frac{\rho^2}{n_i} + \frac{\rho^3}{\Delta} + \beta_i \rho \Delta \right) \\
&\overset{(iii)}{\lesssim} (1-\delta_i/2)\overline{c_{k_i,l_t,t}^2} + \delta_i \overline{\mathbb{E}\|\nabla \bar{f}_i(x_{i,t-1})\|^2} + \frac{\rho}{\Delta}D^2 \overline{\mathbb{E}\|\nabla f_i(x_{i,t-1})\|^2} + \left( \frac{\rho^2}{n_i} + \frac{\rho^3}{\Delta} + \beta_i \rho \Delta \right) \\
&\overset{(iv)}{\lesssim} \overline{(1-\delta_i/2)^{l_t} c_{k_i,1,t}^2} + \delta_i \overline{\mathbb{E}\|\nabla \bar{f}_i(x_{i,t-1})\|^2} + \frac{\rho}{\Delta}D^2 \overline{\mathbb{E}\|\nabla f_i(x_{i,t-1})\|^2} + \left( \frac{\rho^2}{n_i} + \frac{\rho^3}{\Delta} + \beta_i \rho \Delta \right) \\
&\overset{(v)}{\leq} \frac{\rho}{\Delta} \overline{c_{k_i,1,t}^2} + \delta_i \overline{\mathbb{E}\|\nabla \bar{f}_i(x_{i,t-1})\|^2} + \frac{\rho}{\Delta}D^2 \overline{\mathbb{E}\|\nabla f_i(x_{i,t-1})\|^2} + \left( \frac{\rho^2}{n_i} + \frac{\rho^3}{\Delta} + \beta_i \rho \Delta \right). \quad (19)
\end{aligned}
$$

Justifications for the labeled steps are:

- (i) Young's inequality: $\|x+y\|^2 \leq (1+\epsilon)x^2 + (1+{}^1\!/\!_\epsilon)y^2$ for any $\epsilon > 0$.

- (ii) We prove this bound in Lemmas 8 and 12. Importantly, it shows that the clustering error is composed of two quantities: $\mathcal{E}_1$, the error contributed by good points from the cluster's perspective, and $\mathcal{E}_2$, the error contributed by the bad points from the cluster's perspective.

- (iii) Assumption that $\beta_i \lesssim \min(\delta_i, {}^{\delta_i}\!/\!_{A^4})$

- (iv) Since $\mathbb{E}\|v_{i,l_t,t} - \nabla \bar{f}_i(x_{i,t-1})\|^2 = c_{k_i,l_t+1,t}^2$, the inequality forms a recursion which we unroll over $l_t$ steps.

- (v) Assumption that $l_t \geq \max(1, \frac{\log(\rho/\Delta)}{\log(1-\delta_i/2)})$

Finally, we note that

$$
\begin{aligned}
c_{k_i,1,t}^2 &= \mathbb{E}\|g_i(x_{i,t-1}) - \mathbb{E}_x \bar{g}_i(x_{i,t-1})\|^2 \\
&\lesssim \mathbb{E}\|g_i(x_{i,t-1}) - \mathbb{E}_x g_i(x_{i,t-1})\|^2 + \mathbb{E}\|\mathbb{E}_x g_i(x_{i,t-1}) - \mathbb{E}_x \bar{g}_i(x_{i,t-1})\|^2 \\
&\leq \rho^2 + A^2 \mathbb{E}\|\nabla \bar{f}_i(x_{i,t-1})\|^2.
\end{aligned}
$$

Applying this bound to (19), and applying the bound on $\Delta$ from the theorem statement, we have

$$
\begin{aligned}
&\overline{\mathbb{E}\|v_{i,l_t,t} - \nabla \bar{f}_i(x_{i,t-1})\|^2} \\
&\lesssim \left(\delta_i + \frac{\rho A^2}{\Delta}\right) \overline{\mathbb{E}\|\nabla \bar{f}_i(x_{i,t-1})\|^2} + \frac{\rho}{\Delta} D^2 \overline{\mathbb{E}\|\nabla f_i(x_{i,t-1})\|^2} + \left(\frac{\rho^2}{n_i} + \frac{\rho^3}{\Delta} + \beta_i \rho \Delta\right) \\
&\lesssim \delta_i \overline{\mathbb{E}\|\nabla \bar{f}_i(x_{i,t-1})\|^2} + \frac{\rho}{\Delta} D^2 \overline{\mathbb{E}\|\nabla f_i(x_{i,t-1})\|^2} + \left(\frac{\rho^2}{n_i} + \frac{\rho^3}{\Delta} + \beta_i \rho \Delta\right).
\end{aligned}
$$

$\square$

**Lemma 8** (Clustering Error due to Good Points)**.**

$$
\begin{aligned}
&\overline{\mathbb{E}\left\|\left(\frac{1}{|\mathcal{G}_i|} \sum_{j \in \mathcal{G}_i} y_{j,l_t,t}\right) - \mathbb{E}_x \bar{g}_i(x_{i,t-1})\right\|^2} \\
&\lesssim (1-\delta_i)\overline{c_{k_i,l_t,t}^2} + \delta_i \overline{\mathbb{E}\|\nabla \bar{f}_i(x_{i,t-1})\|^2} + \frac{\rho}{\Delta} D^2 \overline{\mathbb{E}\|\nabla f_i(x_{i,t-1})\|^2} + \left(\frac{\rho^2}{n_i} + \frac{\rho^3}{\Delta}\right)
\end{aligned}
$$

*Proof of Lemma 8.* We prove the main result in the sequence of inequalities below and then justify the labeled steps.

$$\mathbb{E}\left\|\left(\frac{1}{|\mathcal{G}_i|}\sum_{j\in\mathcal{G}_i}y_{j,l_t,t}\right)-\mathbb{E}_x\bar{g}_i(x_{i,t-1})\right\|^2$$

$$=\mathbb{E}\left\|\left(\frac{1}{|\mathcal{G}_i|}\sum_{j\in\mathcal{G}_i}y_{j,l_t,t}\right)-\frac{1}{|\mathcal{G}_i|}\sum_{j\in\mathcal{G}_i:j\sim i}\mathbb{E}_x g_j(x_{i,t-1})-\left(\frac{1}{n_i}-\frac{1}{|\mathcal{G}_i|}\right)\sum_{j\in\mathcal{G}_i:j\sim i}\mathbb{E}_x g_j(x_{i,t-1})\right\|^2$$

$$=\mathbb{E}\left\|\left(\frac{1}{|\mathcal{G}_i|}\sum_{j\in\mathcal{G}_i:j\sim i}(y_{j,l_t,t}-\mathbb{E}_x g_j(x_{i,t-1}))\right)+\left(\frac{1}{|\mathcal{G}_i|}\sum_{j\in\mathcal{G}_i:j\not\sim i}(y_{j,l_t,t}-\mathbb{E}_x\bar{g}_i(x_{i,t-1}))\right)\right\|^2$$

$$\overset{(i)}{\leq}\left(1+\frac{2}{\delta_i}\right)\mathbb{E}\left\|\frac{1}{|\mathcal{G}_i|}\sum_{j\in\mathcal{G}_i:j\sim i}(y_{j,l_t,t}-\mathbb{E}_x g_j(x_{i,t-1}))\right\|^2+\left(1+\frac{\delta_i}{2}\right)\mathbb{E}\left\|\frac{1}{|\mathcal{G}_i|}\sum_{j\in\mathcal{G}_i:j\not\sim i}(y_{j,l_t,t}-\mathbb{E}_x\bar{g}_i(x_{i,t-1}))\right\|^2$$

$$\overset{(ii)}{\lesssim}\left(1+\frac{2}{\delta_i}\right)\mathbb{E}\left\|\frac{1}{|\mathcal{G}_i|}\sum_{j\in\mathcal{G}_i:j\sim i}(\mathbb{E}y_{j,l_t,t}-\mathbb{E}_x g_j(x_{i,t-1}))\right\|^2+\left(1+\frac{2}{\delta_i}\right)\mathbb{E}\left\|\frac{1}{|\mathcal{G}_i|}\sum_{j\in\mathcal{G}_i:j\sim i}(y_{j,l_t,t}-\mathbb{E}y_{j,l_t,t})\right\|^2$$

$$\quad+\left(1+\frac{\delta_i}{2}\right)\mathbb{E}\left\|\frac{1}{|\mathcal{G}_i|}\sum_{j\in\mathcal{G}_i:j\not\sim i}(y_{j,l_t,t}-\mathbb{E}_x\bar{g}_i(x_{i,t-1}))\right\|^2$$

$$\overset{(iii)}{\lesssim}\left(1+\frac{2}{\delta_i}\right)\mathbb{E}\left\|\frac{1}{|\mathcal{G}_i|}\sum_{j\in\mathcal{G}_i:j\sim i}\mathbb{E}(y_{j,l_t,t}-g_j(x_{i,t-1}))\right\|^2+\left(1+\frac{2}{\delta_i}\right)\mathbb{E}\left\|\frac{1}{|\mathcal{G}_i|}\sum_{j\in\mathcal{G}_i:j\sim i}(\mathbb{E}_x g_j(x_{i,t-1})-\mathbb{E}g_j(x_{i,t-1}))\right\|^2$$

$$\quad+\left(1+\frac{2}{\delta_i}\right)\mathbb{E}\left\|\frac{1}{|\mathcal{G}_i|}\sum_{j\in\mathcal{G}_i:j\sim i}(y_{j,l_t,t}-\mathbb{E}y_{j,l_t,t})\right\|^2+\left(1+\frac{\delta_i}{2}\right)\mathbb{E}\left\|\frac{1}{|\mathcal{G}_i|}\sum_{j\in\mathcal{G}_i:j\not\sim i}(y_{j,l_t,t}-\mathbb{E}_x\bar{g}_i(x_{i,t-1}))\right\|^2$$

$$\overset{(iv)}{\lesssim}\left(1+\frac{2}{\delta_i}\right)\delta_i^2\underbrace{\mathbb{E}_{j\in\mathcal{G}_i:j\sim i}\|\mathbb{E}(y_{j,l_t,t}-g_j(x_{i,t-1}))\|^2}_{\mathcal{T}_1}+\left(1+\frac{2}{\delta_i}\right)\frac{n_i}{|\mathcal{G}_i|^2}\rho^2$$

$$\quad+\left(1+\frac{2}{\delta_i}\right)\underbrace{\mathbb{E}\left\|\frac{1}{|\mathcal{G}_i|}\sum_{j\in\mathcal{G}_i:j\sim i}(y_{j,l_t,t}-\mathbb{E}y_{j,l_t,t})\right\|^2}_{\mathcal{T}_2}+\left(1+\frac{\delta_i}{2}\right)(1-\delta_i)^2\underbrace{\mathbb{E}_{j\in\mathcal{G}_i:j\not\sim i}\|y_{j,l_t,t}-\mathbb{E}_x\bar{g}_i(x_{i,t-1})\|^2}_{\mathcal{T}_3}$$

$$\overset{(v)}{\lesssim}\delta_i\left(c_{k_i,l_t,t}^2+\mathbb{E}\|\nabla\bar{f}_i(x_{i,t-1})\|^2+\frac{\rho^3}{\delta_i\Delta}\right)+\left(1+\frac{2}{\delta_i}\right)\frac{n_i}{|\mathcal{G}_i|^2}\rho^2$$

$$\quad+\left(1+\frac{\delta_i}{2}\right)(1-\delta_i)^2\left(\left(1+\frac{\delta_i}{2}\right)c_{k_i,l_t,t}^2+\delta_i\mathbb{E}\|\nabla\bar{f}_i(x_{i,t-1})\|^2+\frac{\rho}{\Delta}D^2\mathbb{E}\|\nabla f_i(x_{i,t-1})\|^2+\frac{\rho^3}{\Delta}\right)$$

$$\overset{(vi)}{\lesssim}\delta_i\left(c_{k_i,l_t,t}^2+\mathbb{E}\|\nabla\bar{f}_i(x_{i,t-1})\|^2+\frac{\rho^3}{\delta_i\Delta}\right)+\frac{\rho^2}{n_i}$$

$$\quad+(1-\delta_i)\left(c_{k_i,l_t,t}^2+\delta_i\mathbb{E}\|\nabla\bar{f}_i(x_{i,t-1})\|^2+\frac{\rho}{\Delta}D^2\mathbb{E}\|\nabla f_i(x_{i,t-1})\|^2+\frac{\rho^3}{\Delta}\right)$$

$$\lesssim(1-\delta_i)c_{k_i,l_t,t}^2+\delta_i\mathbb{E}\|\nabla\bar{f}_i(x_{i,t-1})\|^2+\frac{\rho}{\Delta}D^2\mathbb{E}\|\nabla f_i(x_{i,t-1})\|^2+\left(\frac{\rho^2}{n_i}+\frac{\rho^3}{\Delta}\right).$$

Justifications for the labeled steps are:

- (i),(ii),(iii) Young's inequality

- (iv) First, we can can interchange the sum and the norm due to independent stochasticity of the gradients. Then by the Tower Property and Law of Total Variance for the 1st and 3rd steps

respectively,

$$
\begin{aligned}
\mathbb{E}\|\mathbb{E}_x g_j(x_{i,t-1}) - \mathbb{E}g_j(x_{i,t-1}))\|^2 &= \mathbb{E}\|\mathbb{E}_x g_j(x_{i,t-1}) - \mathbb{E}[\mathbb{E}_x g_j(x_{i,t-1})]\|^2 \\
&= \mathrm{Var}(\mathbb{E}_x(g_j(x_{i,t-1}))) \\
&= \mathrm{Var}(g_j(x_{i,t-1})) - \mathbb{E}(\mathrm{Var}_x(g_j(x_{i,t-1}))) \\
&\leq \mathrm{Var}(g_j(x_{i,t-1})) - \mathbb{E}\|g_j(x_{i,t-1}) - \mathbb{E}_x g_j(x_{i,t-1})\|^2 \\
&\lesssim \rho^2,
\end{aligned}
$$

where the last inequality follows since the two terms above it are both bounded by $\rho^2$.

- (v) We prove this bound in Lemmas 9, 10, and 11. It shows that, from point $i$'s perspective, the error of its cluster-center-estimate is composed of three quantities: $\mathcal{T}_1$, the error introduced by our thresholding procedure on the good points which belong to $i$'s cluster (and therefore ideally are included within the thresholding radius); $\mathcal{T}_2$, which accounts for the variance of the clipped points in $i$'s cluster; and $\mathcal{T}_3$, the error due to the good points which don't belong to $i$'s cluster (and therefore ideally are forced outside the thresholding radius).

- (vi) $(1 + {}^x\!/_2)^2 (1-x)^2 \leq 1 - x$ for all $x \in [0,1]$

$\square$

**Lemma 9** (Bound $\mathcal{T}_1$: Error due to In-Cluster Good Points).

$$
\overline{\mathbb{E}_{j \in \mathcal{G}_i : j \sim i} \|\mathbb{E}(y_{j,l_t,t} - g_j(x_{i,t-1}))\|^2} \lesssim \overline{c_{k_i,l_t,t}^2} + \overline{\mathbb{E}\|\nabla \bar{f}_i(x_{i,t-1})\|^2} + \frac{\rho^3}{\delta_i \Delta}.
$$

*Proof of Lemma 9.* In this sequence of steps, we bound the clustering error due to good points from client $i$'s cluster. By definition of $y_{j,l_t,t}$,

$$
\begin{aligned}
\mathbb{E}\|y_{j,l_t,t} - g_j(x_{i,t-1})\| &= \mathbb{E}[\|v_{i,l_t-1,t} - g_j(x_{i,t-1})\| \mathbb{1}(\|v_{i,l_t-1,t} - g_j(x_{i,t-1})\| > \tau_{k_i,l_t,t})] \\
&\leq \frac{\mathbb{E}[\|v_{i,l_t-1,t} - g_j(x_{i,t-1})\|^2 \mathbb{1}(\|v_{i,l_t-1,t} - g_j(x_{i,t-1})\| > \tau_{k_i,l_t,t})]}{\tau_{k_i^t,l_t}} \\
&\leq \frac{\mathbb{E}\|v_{i,l_t-1,t} - g_j(x_{i,t-1})\|^2}{\tau_{k_i,l_t,t}}.
\end{aligned}
$$

Therefore, by Jensen's inequality and plugging in the value for $\tau_{k_i,l_t,t}$,

$$
\begin{aligned}
&\|\mathbb{E}(y_{j,l_t,t} - g_j(x_{i,t-1}))\|^2 \\
&\leq (\mathbb{E}\|y_{j,l_t,t} - g_j(x_{i,t-1})\|)^2 \\
&\leq \frac{(\mathbb{E}\|v_{i,l_t-1,t} - g_j(x_{i,t-1})\|^2)^2}{\tau_{k_i,l_t,t}^2} \\
&\lesssim \frac{(\mathbb{E}\|v_{i,l_t-1,t} - \mathbb{E}_x \bar{g}_i(x_{i,t-1})\|^2 + \mathbb{E}\|\mathbb{E}_x \bar{g}_i(x_{i,t-1}) - \mathbb{E}_x g_j(x_{i,t-1})\|^2 + \mathbb{E}\|g_j(x_{i,t-1}) - \mathbb{E}_x g_j(x_{i,t-1})\|^2)^2}{\tau_{k_i,l_t,t}^2} \\
&\leq \frac{(c_{k_i,l_t,t}^2 + A^2 \mathbb{E}\|\nabla \bar{f}_i(x_{i,t-1})\|^2 + \rho^2)^2}{\tau_{k_i,l_t,t}^2} \\
&\lesssim \frac{(c_{k_i,l_t,t}^2 + A^2 \mathbb{E}\|\nabla \bar{f}_i(x_{i,t-1})\|^2 + \rho^2)^2}{c_{k_i,l_t,t}^2 + A^4 \mathbb{E}\|\nabla \bar{f}_i(x_{i,t-1})\|^2 + \delta_i \rho \Delta} \\
&\lesssim \left(1 + \frac{\rho}{\delta_i \Delta}\right) c_{k_i,l_t,t}^2 + \left(1 + \frac{\rho A^2}{\delta_i \Delta}\right) \mathbb{E}\|\nabla \bar{f}_i(x_{i,t-1})\|^2 + \frac{\rho^3}{\delta_i \Delta} \\
&\lesssim c_{k_i,l_t,t}^2 + \mathbb{E}\|\nabla \bar{f}_i(x_{i,t-1})\|^2 + \frac{\rho^3}{\delta_i \Delta}, \quad (20)
\end{aligned}
$$

where the last inequality follows from constraints on $\Delta$. The second inequality follows from Young's inequality. The second-to-last inequality follows by separating the fraction into a sum of fractions and selecting terms from the denominator for each fraction that cancel with terms in the numerator to achieve the desired rate.

Summing (20) over $t$ and dividing by $T$, we have

$$\overline{\mathbb{E}_{j\in\mathcal{G}_i:j\sim i}\|\mathbb{E}_x(y_{j,l_t,t} - g_j(x_{i,t-1}))\|^2} = \overline{\mathbb{E}\|y_{j,l_t,t} - g_j(x_{i,t-1})\|^2}$$
$$\lesssim \overline{c_{k_i,l_t,t}^2} + \overline{\mathbb{E}\|\nabla\bar{f}_i(x_{i,t-1})\|^2} + \frac{\rho^3}{\delta_i\Delta}.$$

$\square$

**Lemma 10** (Bound $\mathcal{T}_2$: Variance of Clipped Points)**.**

$$\mathbb{E}\left\|\frac{1}{|\mathcal{G}_i|}\sum_{j\in\mathcal{G}_i:j\sim i}(y_{j,l_t,t} - \mathbb{E}y_{j,l_t,t})\right\|^2 \leq \frac{n_i}{|\mathcal{G}_i|^2}\rho^2.$$

*Proof of Lemma 10.* The first thing to note is that the elements in the sum $\sum_{j\in\mathcal{G}_i:j\sim i}(y_{j,l_t,t} - \mathbb{E}y_{j,l_t,t})$ are not independent. Therefore, we cannot get rid of the cross terms when expanding the squared-norm. However, if for each round of thresholding we sampled a fresh batch of points to set the new cluster-center estimate, then the terms would be independent. With such a resampling strategy, our bounds in these proofs only change by a constant factor. Therefore, for ease of analysis, we will assume the terms in the sum are independent. In that case,

$$\mathbb{E}\left\|\frac{1}{|\mathcal{G}_i|}\sum_{j\in\mathcal{G}_i:j\sim i}(y_{j,l_t,t} - \mathbb{E}y_{j,l_t,t})\right\|^2 \leq \frac{n_i}{|\mathcal{G}_i|^2}\mathbb{E}\|y_{j,l_t,t} - \mathbb{E}y_{j,l_t,t}\|^2$$
$$\leq \frac{n_i}{|\mathcal{G}_i|^2}\mathbb{E}\|g_j(x_{i,t-1}) - \mathbb{E}g_j(x_{i,t-1})\|^2$$
$$\leq \frac{n_i}{|\mathcal{G}_i|^2}\rho^2,$$

where the second-to-last inequality follows from the contractivity of the thresholding procedure. $\square$

**Lemma 11** (Bound $\mathcal{T}_3$: Error due to Out-of-Cluster Good Points)**.**

$$\overline{\mathbb{E}_{j\in\mathcal{G}_i:j\nsim i}\|y_{j,l_t,t} - \mathbb{E}_x\bar{g}_i(x_{i,t-1})\|^2} \lesssim \left(1 + \frac{\delta_i}{2}\right)\overline{c_{k_i,l_t,t}^2} + \delta_i\overline{\mathbb{E}\|\nabla\bar{f}_i(x_{i,t-1})\|^2} + \frac{\rho}{\Delta}D^2\overline{\mathbb{E}\|\nabla f_i(x_{i,t-1})\|^2} + \frac{\rho^3}{\Delta}.$$

*Proof of Lemma 11.* This sequence of steps bounds the clustering error due to points not from client $i$'s cluster. Using Young's inequality for the first step,

$$\mathbb{E}_{j\in\mathcal{G}_i:j\nsim i}\|y_{j,l_t,t} - \mathbb{E}_x\bar{g}_i(x_{i,t-1})\|^2$$
$$\leq \left(1 + \frac{\delta_i}{2}\right)\mathbb{E}\|v_{i,l_t-1,t} - \mathbb{E}_x\bar{g}_i(x_{i,t-1})\|^2 + \left(1 + \frac{2}{\delta_i}\right)\mathbb{E}_{j\in\mathcal{G}_i:j\nsim i}\|y_{j,l_t,t} - v_{i,l_t-1,t}\|^2$$
$$\leq \left(1 + \frac{\delta_i}{2}\right)c_{k_i,l_t,t}^2 + \left(1 + \frac{2}{\delta_i}\right)\mathbb{E}_{j\in\mathcal{G}_i:j\nsim i}\|y_{j,l_t,t} - v_{i,l_t-1,t}\|^2$$
$$= \left(1 + \frac{\delta_i}{2}\right)c_{k_i,l_t,t}^2 + \left(1 + \frac{2}{\delta_i}\right)\mathbb{E}_{j\in\mathcal{G}_i:j\nsim i}[\|g_j(x_{i,t-1}) - v_{i,l_t-1,t}\|^2\mathbb{1}\{\|g_j(x_{i,t-1}) - v_{i,l_t-1,t}\| \leq \tau_{k_i,l_t,t}\}]$$
$$\leq \left(1 + \frac{\delta_i}{2}\right)c_{k_i,l_t,t}^2 + \left(1 + \frac{2}{\delta_i}\right)\tau_{k_i,l_t,t}^2\mathbb{P}_{j\in\mathcal{G}_i:j\nsim i}(\|g_j(x_{i,t-1}) - v_{i,l_t-1,t}\| \leq \tau_{k_i,l_t,t})$$
$$\lesssim \left(1 + \frac{\delta_i}{2}\right)c_{k_i,l_t,t}^2 + \left(\frac{1}{\delta_i}\right)(c_{k_i,l_t,t}^2 + A^4\mathbb{E}\|\nabla\bar{f}_i(x_{i,t-1})\|^2 + \delta_i\rho\Delta)\mathbb{P}_{j\in\mathcal{G}_i:j\nsim i}(\|g_j(x_{i,t-1}) - v_{i,l_t-1,t}\| \leq \tau_{k_i,l_t,t}).$$

The next step is to bound the probability term in the inequality above. Note that if $\|v_{i,l_t-1,t} - g_j(x_{i,t-1})\| \leq \tau_{k_i^t,l_t}$, then

$$\|\mathbb{E}_x g_j(x_{i,t-1}) - \mathbb{E}_x g_i(x_{i,t-1})\|^2 \lesssim \|g_j(x_{i,t-1}) - \mathbb{E}_x g_j(x_{i,t-1})\|^2 + \|g_j(x_{i,t-1}) - v_{i,l_t-1,t}\|^2 + \|v_{i,l_t-1,t} - \mathbb{E}_x g_i(x_{i,t-1})\|^2$$

$$\begin{aligned}
&\lesssim \|g_j(x_{i,t-1}) - \mathbb{E}_x g_j(x_{i,t-1})\|^2 + \tau_{k_i,l_t,t}^2 \\
&\quad + \|v_{i,l_t-1,t} - \mathbb{E}_x \bar{g}_i(x_{i,t-1})\|^2 + \|\mathbb{E}_x g_i(x_{i,t-1}) - \mathbb{E}_x \bar{g}_i(x_{i,t-1})\|^2 \\
&\lesssim \|g_j(x_{i,t-1}) - \mathbb{E}_x g_j(x_{i,t-1})\|^2 + \tau_{k_i,l_t,t}^2 \\
&\quad + \|v_{i,l_t-1,t} - \mathbb{E}_x \bar{g}_i(x_{i,t-1})\|^2 + A^2 \|\nabla \bar{f}_i(x_{i,t-1})\|^2.
\end{aligned}$$

By Assumption 4, the previous inequality implies

$$\Delta^2 - D^2 \|\nabla f_i(x_{i,t-1})\|^2 \lesssim \|g_j(x_{i,t-1}) - \mathbb{E}_x g_j(x_{i,t-1})\|^2 + \tau_{k_i,l_t,t}^2 + A^2 \|\nabla \bar{f}_i(x_{i,t-1})\|^2 + \|v_{i,l_t-1,t} - \mathbb{E}_x \bar{g}_i(x_{i,t-1})\|^2$$

which, summing over $t$ and dividing by $T$, implies

$$\overline{\|g_j(x_{i,t-1}) - \mathbb{E}_x g_j(x_{i,t-1})\|^2} + \overline{\|v_{i,l_t-1,t} - \mathbb{E}_x \bar{g}_i(x_{i,t-1})\|^2} + A^2 \overline{\|\nabla \bar{f}_i(x_{i,t-1})\|^2} + D^2 \overline{\|\nabla f_i(x_{i,t-1})\|^2} \gtrsim \Delta^2 - \overline{\tau_{k_i,l_t,t}^2}.$$

By Markov's inequality, the probability of this event is upper-bounded by

$$\frac{\rho^2 + \overline{\mathbb{E}\|v_{i,l_t-1,t} - \mathbb{E}_x \bar{g}_i(x_{i,t-1})\|^2} + A^2 \overline{\mathbb{E}\|\nabla \bar{f}_i(x_{i,t-1})\|^2} + D^2 \overline{\mathbb{E}\|\nabla f_i(x_{i,t-1})\|^2}}{\Delta^2 - \overline{\tau_{k_i,l_t,t}^2}}$$

$$\lesssim \frac{\rho^2 + \overline{c_{k_i,l_t,t}^2} + A^2 \overline{\mathbb{E}\|\nabla \bar{f}_i(x_{i,t-1})\|^2} + D^2 \overline{\mathbb{E}\|\nabla f_i(x_{i,t-1})\|^2}}{\Delta^2},$$

where the second inequality holds due to the constraint on $\Delta$ from the theorem statement. Therefore,

$$\begin{aligned}
&\overline{\mathbb{E}_{j \in \mathcal{G}_i : j \nsim i} \|y_{j,l_t,t} - \mathbb{E}_x \bar{g}_i(x_{i,t-1})\|^2} \\
&\lesssim \left(1 + \frac{\delta_i}{2} + \frac{\rho}{\Delta} + \frac{\rho^2 + \overline{c_{k_i,l_t,t}^2} + A^2 \overline{\mathbb{E}\|\nabla \bar{f}_i(x_{i,t-1})\|^2} + D^2 \overline{\mathbb{E}\|\nabla f_i(x_{i,t-1})\|^2}}{\delta_i \Delta^2}\right) \overline{c_{k_i,l_t,t}^2} \\
&\quad + \left(\frac{\rho A^2}{\Delta} + \frac{A^4(\rho^2 + \overline{c_{k_i,l_t,t}^2} + A^2 \overline{\mathbb{E}\|\nabla \bar{f}_i(x_{i,t-1})\|^2} + D^2 \overline{\mathbb{E}\|\nabla f_i(x_{i,t-1})\|^2})}{\delta_i \Delta^2}\right) \overline{\mathbb{E}\|\nabla \bar{f}_i(x_{i,t-1})\|^2} \\
&\quad + \frac{\rho}{\Delta} D^2 \overline{\mathbb{E}\|\nabla f_i(x_{i,t-1})\|^2} + \frac{\rho^3}{\Delta} \\
&\lesssim \left(1 + \frac{\delta_i}{2}\right) \overline{c_{k_i,l_t,t}^2} + \delta_i \overline{\mathbb{E}\|\nabla \bar{f}_i(x_{i,t-1})\|^2} + \frac{\rho}{\Delta} D^2 \overline{\mathbb{E}\|\nabla f_i(x_{i,t-1})\|^2} + \frac{\rho^3}{\Delta},
\end{aligned}$$

where for the last inequality we again apply the constraint on $\Delta$. $\qquad \square$

**Lemma 12** (Clustering Error due to Bad Points).

$$\overline{\mathbb{E}\left\|\left(\frac{1}{|\mathcal{B}_i|} \sum_{j \in \mathcal{B}_i} y_{j,l_t,t}\right) - \mathbb{E}_x \bar{g}_i(x_{i,t-1})\right\|^2} \lesssim \overline{c_{k_i,l_t,t}^2} + A^4 \overline{\mathbb{E}\|\nabla \bar{f}_i(x_{i,t-1})\|^2} + \delta_i \rho \Delta$$

*Proof of Lemma 12.* This lemma bounds the clustering error due to the bad clients from client $i$'s perspective. The goal of such clients would be to corrupt the cluster-center estimate of client $i$'s cluster as much as possible at each round. They can have the maximum negative effect by setting their gradients to be just inside the thresholding radius around client $i$'s cluster-center estimate. This way, the gradients will keep their value (rather than be assigned the value of the current cluster-center estimate per our update rule), but they will

have maximal effect in moving the cluster-center estimate from its current position. Therefore, in step 3 of the inequalities below, we can not do better than bounding the distance between these bad points and the current cluster center estimate (i.e. $\|y_{j,l_t,t} - v_{i,l_t-1,t}\|^2$) by the thresholding radius ($\tau^2_{k_i,l_t,t}$).

$$
\begin{aligned}
\mathbb{E}\left\|\left(\frac{1}{|\mathcal{B}_i|}\sum_{j\in\mathcal{B}_i} y_{j,l_t,t}\right) - \mathbb{E}_x \bar{g}_i(x_{i,t-1})\right\|^2 &\leq \mathbb{E}_{j\in\mathcal{B}_i}\|y_{j,l_t,t} - \mathbb{E}_x \bar{g}_i(x_{i,t-1})\|^2 \\
&\lesssim \mathbb{E}_{j\in\mathcal{B}_i}\|y_{j,l_t,t} - v_{i,l_t-1,t}\|^2 + \mathbb{E}\|v_{i,l_t-1,t} - \mathbb{E}_x \bar{g}_i(x_{i,t-1})\|^2 \\
&\leq \tau^2_{k_i,l_t,t} + c^2_{k_i,l_t,t} \\
&\lesssim c^2_{k_i,l_t,t} + A^4 \mathbb{E}\|\nabla \bar{f}_i(x_{i,t-1})\|^2 + \delta_i \rho \Delta.
\end{aligned}
$$

The last inequality applies the definition of $\tau_{k_i,l_t,t}$, and the result of the lemma follows by summing this inequality over $t$ and dividing by $T$. □

## A.4 Proof of Corollary 1

*Proof.* We will show that there exist $A$, $D$, and $\Delta$ for which Assumptions 4 and 5 are satisfied for smooth and strongly-convex losses. Then the result follows directly from Theorem 3.

Note that for $h$ an $L$-smooth function and $g$ a $\mu$-strongly-convex function with shared optimum $x^*$, the following inequality holds:

$$
\|\nabla h(x)\|^2 \leq \left(\frac{L}{\mu}\right)^2 \|\nabla g(x)\|^2. \tag{21}
$$

To see this, note that by $L$-smoothness of $h$

$$
\|\nabla h(x)\|^2 = \|\nabla h(x) - \nabla h(x^*)\|^2 \leq L^2\|x - x^*\|^2. \tag{22}
$$

By $\mu$-strong-convexity of $g$ and Cauchy-Schwarz inequality,

$$
\mu\|x - x^*\|^2 \leq \langle \nabla g(x) - \nabla g(x^*), x - x^* \rangle \leq \|\nabla g(x)\|\|x - x^*\|. \tag{23}
$$

Rearranging terms in (23), squaring both sides, and combining it with (22) gives (21).

We can now apply (21) to show that Assumptions 4 and 5 hold.

For Assumption 4, let $h(x) = f_i(x) - \nabla \bar{f}_i(x)$ and $g(x) = \nabla \bar{f}_i(x)$. Thus $h$ and $g$ have the same optimum. Since the average of $\mu$-strongly-convex functions is $\mu$-strongly-convex, $g$ is $\mu$-strongly-convex. By $L$-smoothness of $f_i$,

$$
\begin{aligned}
\|(\nabla f_i(x) - \nabla \bar{f}_i(x)) - (\nabla f_i(y) - \nabla \bar{f}_i(y))\| &\leq \|\nabla f_i(x) - \nabla f_i(y)\| + \|\nabla \bar{f}_i(x)) - \nabla \bar{f}_i(y)\| \\
&\leq 2L\|x - y\|,
\end{aligned}
$$

showing that $h$ is $2L$-smooth. Therefore, by (21)

$$
\|\nabla f_i(x) - \nabla \bar{f}_i(x)\|^2 \leq \left(\frac{2L}{\mu}\right)^2 \|\nabla \bar{f}_i(x)\|^2,
$$

which shows that Assumption 4 is satisfied with $A = 2L/\mu$.

For Assumption 5, let $x_i^*$ be client $i$'s optimum (equivalently the optimum of all clients in client $i$'s cluster).

$$
\begin{aligned}
\|\nabla f_i(x) - \nabla f_j(x)\|^2 &= \|\nabla f_i(x) - (\nabla f_j(x) - \nabla f_j(x_i^*)) - (\nabla f_j(x_i^*) - \nabla f_i(x_i^*)))\|^2 \\
&\overset{(i)}{\geq} \frac{1}{2}\|\nabla f_j(x_i^*) - \nabla f_i(x_i^*)\|^2 - \|\nabla f_i(x) - (\nabla f_j(x) - \nabla f_j(x_i^*))\|^2 \\
&\overset{(ii)}{\geq} \frac{1}{2}\|\nabla f_j(x_i^*) - \nabla f_i(x_i^*)\|^2 - 2\|\nabla f_i(x)\|^2 - 2\|\nabla f_j(x) - \nabla f_j(x_i^*)\|^2 \\
&\overset{(iii)}{\geq} \frac{1}{2}\|\nabla f_j(x_i^*) - \nabla f_i(x_i^*)\|^2 - 2\|\nabla f_i(x)\|^2 - 2(L/\mu)^2\|\nabla f_i(x)\|^2 \\
&= \frac{1}{2}\|\nabla f_j(x_i^*) - \nabla f_i(x_i^*)\|^2 - 2(1 + (L/\mu)^2)\|\nabla f_i(x)\|^2 \\
&= \frac{1}{2}\|\nabla f_j(x_i^*)\|^2 - 2(1 + (L/\mu)^2)\|\nabla f_i(x)\|^2,
\end{aligned}
\tag{24}
$$

where justification for the steps are:

- (i) For all $a$, $b$, it holds that $(a - b)^2 \geq \frac{1}{2}b^2 - a^2$, since this inequality can be rearranged to state $(b/\sqrt{2} - \sqrt{2}a)^2 \geq 0$.

- (ii) Young's inequality.

- (iii) Set $h(x) = f_j(x) - \langle f_j(x_i^*), x\rangle$ and $g(x) = f_i(x)$. Then $\nabla h(x) = \nabla f_j(x) - \nabla f_j(x_i^*)$, from which we see that $h$ and $g$ have the same optimum $x_i^*$ and $h$ is $L$-smooth (since $\|\nabla h(x) - \nabla h(y)\| = \|(\nabla f_j(x) - \nabla f_j(x_i^*)) - (\nabla f_j(y) - \nabla f_j(x_i^*))\| \leq L\|x - y\|$). Applying (21) gives the desired result.

Therefore (24) shows that Assumption 5 is satisfied with $\Delta = \max_{j \nsim i} \frac{1}{\sqrt{2}}\|\nabla f_j(x_i^*)\|$, and $D = \sqrt{2(1 + (L/\mu)^2)}$.

Applying these values for $A$, $D$, and $\Delta$ to Theorem 3 completes the proof. $\qquad\square$

### A.5 Proof of Theorem 4

First we establish some notation.

**Notation.**

- $\mathcal{G}_i$ are the good clients and $\mathcal{B}_i$ the bad clients from client $i$'s perspective.

- $\mathbb{E}_x$ denotes conditional expectation given the parameter, e.g. $\mathbb{E}_x g(x) = \mathbb{E}[g(x)|x]$. $\mathbb{E}$ denotes expectation over all randomness.

- $k_i^t$ is the cluster to which client $i$ is assigned at round $t$ of the algorithm.

- $\overline{X_t} \triangleq \frac{1}{T}\sum_{t=1}^{T} X_t$ for a general variable $X_t$ indexed by $t$.

- $\bar{f}_i(x) \triangleq \frac{1}{n_i}\sum_{j \in \mathcal{G}_i : j \sim i} f_j(x)$

- $\bar{m}_{i,t} \triangleq \frac{1}{n_i}\sum_{j \in \mathcal{G}_i : j \sim i} m_{j,t}$

- We introduce a variable $\rho^2$ to bound the variance of the momentums

$$
\mathbb{E}_x\|m_{i,t} - \mathbb{E}_x m_{i,t}\|^2 \leq \rho^2,
$$

  and show in Lemma 13 how this can be written in terms of the variance of the gradients, $\sigma^2$.

- $l_t$ is the number of rounds that Threshold-Clustering is run in round $t$ of Federated-Clustering.

- $k_i$ denotes the cluster to which client $i$ is assigned.

- $v_{k_i,l,t}$ denotes the gradient update for client $i$ in round $t$ of Momentum-Clustering and round $l$ of Threshold-Clustering. That is, $v_{k_i,l,t}$ corresponds to the quantity returned in Step 4 of Algorithm 4.

- To facilitate the proof, we introduce a variable $c_{k_i,l,t}$ that quantifies the distance from the cluster-center-estimates to the true cluster means. Specifically, for client $i$'s cluster $k_i$ at round $t$ of Federated-Clustering and round $l$ of Threshold-Clustering we set

$$c^2_{k_i,l,t} = \mathbb{E}\|v_{k_i,l-1,t} - \mathbb{E}_x \bar{m}_{i,t}\|^2.$$

- For client $i$'s cluster $k_i$ at round $t$ of Federated-Clustering and round $l$ of Threshold-Clustering, we use thresholding radius

$$\tau^2_{k_i,l,t} \approx c^2_{k_i,l,t} + \delta_i \rho \Delta.$$

- Finally, we introduce a variable, $y_{j,l,t}$, to denote the points clipped by Threshold-Clustering:

$$v_{k_i,l,t} = \frac{1}{N} \sum_{j \in [N]} \underbrace{\mathbb{1}(\|m_{j,t} - v_{k_i,l-1,t}\| \leq \tau_{k_i,l,t}) + v_{k_i,l-1,t} \mathbb{1}(\|m_{j,t} - v_{k_i,l-1,t}\| > \tau_{k_i,l,t})}_{y_{j,l,t}}.$$

*Proof of Theorem 4.* In this proof, our goal is to bound $\overline{\mathbb{E}\|\nabla f_i(x_{i,t-1})\|^2}$. We use $L$-smoothness of the loss objectives to get started, and justify the non-trivial steps afterwards.

$$\mathbb{E}f_i(x_{i,t}) \overset{(i)}{\leq} \mathbb{E}f_i(x_{i,t-1}) + \mathbb{E}\langle \nabla f_i(x_{i,t-1}), x_{i,t} - x_{i,t-1}\rangle + \frac{L}{2}\mathbb{E}\|x_{i,t} - x_{i,t-1}\|^2$$

$$= \mathbb{E}f_i(x_{i,t-1}) - \eta\mathbb{E}\langle \nabla f_i(x_{i,t-1}), v_{k_i,l_t,t}\rangle + \frac{L\eta^2}{2}\mathbb{E}\|v_{k_i,l_t,t}\|^2$$

$$= \mathbb{E}f_i(x_{i,t-1}) + \frac{\eta}{2}\mathbb{E}\|v_{k_i,l_t,t} - \nabla f_i(x_{i,t-1})\|^2 - \frac{\eta}{2}\mathbb{E}\|\nabla f_i(x_{i,t-1})\|^2 - \frac{\eta}{2}(1 - L\eta)\mathbb{E}\|v_{k_i,l_t,t}\|^2$$

$$\overset{(ii)}{\lesssim} \mathbb{E}f_i(x_{i,t-1}) + \eta\mathbb{E}\|v_{k_i,l_t,t} - \mathbb{E}_x \bar{m}_{i,t}\|^2 + \eta\mathbb{E}\|\mathbb{E}_x \bar{m}_{i,t} - \nabla f_i(x_{i,t-1})\|^2$$
$$- \frac{\eta}{2}\mathbb{E}\|\nabla f_i(x_{i,t-1})\|^2 - \frac{\eta}{2}(1 - L\eta)\mathbb{E}\|v_{k_i,l_t,t}\|^2$$

$$\overset{(iii)}{\lesssim} \mathbb{E}f_i(x_{i,t-1}) + \eta\left(\frac{\rho^2}{n_i} + \frac{\rho^3}{\Delta} + \beta_i \rho \Delta\right) + \eta\mathbb{E}\|\mathbb{E}_x \bar{m}_{i,t} - \nabla f_i(x_{i,t-1})\|^2$$
$$- \frac{\eta}{2}\mathbb{E}\|\nabla f_i(x_{i,t-1})\|^2 - \frac{\eta}{2}(1 - L\eta)\mathbb{E}\|v_{k_i,l_t,t}\|^2. \tag{25}$$

Justifications for the labeled steps are:

- (i) $L$-smoothness of $f_i$ and $\eta \leq 1/L$

- (ii) Young's inequality

- (iii) Lemma 14

Now it remains to bound the $\mathbb{E}\|\mathbb{E}_x \bar{m}_{i,t} - \nabla f_i(x_{i,t-1})\|^2$ term.

$$\mathbb{E}\|\mathbb{E}_x \bar{m}_{i,t} - \nabla f_i(x_{i,t-1})\|^2 \leq \mathbb{E}\|\mathbb{E}_x m_{i,t} - \nabla f_i(x_{i,t-1})\|^2$$
$$= (1 - \alpha)^2 \mathbb{E}\|\mathbb{E}_x m_{i,t-1} - \nabla f_i(x_{i,t-2}) + \nabla f_i(x_{i,t-2}) - \nabla f_i(x_{i,t-1})\|^2$$
$$\overset{(i)}{\lesssim} (1 - \alpha)^2(1 + \alpha)\mathbb{E}\|\mathbb{E}_x m_{i,t-1} - \nabla f_i(x_{i,t-2})\|^2 + (1 - \alpha)^2\left(1 + \frac{1}{\alpha}\right)L^2\eta^2\mathbb{E}\|v_{k_i,l_{t-1},t-1}\|^2$$

$$\overset{(ii)}{\leq} \frac{1}{2}(1 - L\eta)\sum_{t=2}^{T}\mathbb{E}\|v_{k_i,l_{t-1},t-1}\|^2,$$

where justifications for the labeled steps are:

- (i) Young's inequality

- (ii) Assumption that $\alpha \gtrsim L\eta$

Plugging this bound back into (25), summing over $t = 1 : T$, and dividing by $T$ gives

$$\frac{\eta}{2T} \sum_{t=1}^{T} \mathbb{E}\|\nabla f_i(x_{i,t-1})\|^2 \lesssim \frac{\mathbb{E}(f_i(x_{i,0}) - f_i^*)}{T} + \eta\left(\frac{\rho^2}{n_i} + \frac{\rho^3}{\Delta} + \beta_i \rho \Delta\right). \tag{26}$$

By the variance reduction from momentum (Lemma 13) it follows from (26) that

$$\frac{1}{T} \sum_{t=1}^{T} \mathbb{E}\|\nabla f_i(x_{i,t-1})\|^2 \lesssim \frac{\mathbb{E}(f_i(x_{i,0}) - f_i^*)}{\eta T} + \left(\frac{\alpha \sigma^2}{n_i} + \frac{\alpha^{3/2} \sigma^3}{\Delta} + \beta_i \sqrt{\alpha} \sigma \Delta\right)$$

$$\lesssim \frac{\mathbb{E}(f_i(x_{i,0}) - f_i^*)}{\eta T} + \left(\frac{L\eta\sigma^2}{n_i} + \frac{L\eta\sigma^3}{\Delta} + \beta_i \sqrt{L\eta} \sigma \Delta\right).$$

Finally, setting $\eta \lesssim \min\left\{\frac{1}{L}, \sqrt{\frac{\mathbb{E}(f_i(x_{i,0}) - f_i^*)}{LT(\sigma^2/n_i + \sigma^3/\Delta)}}\right\}$,

$$\frac{1}{T} \sum_{t=1}^{T} \mathbb{E}\|\nabla f_i(x_{i,t-1})\|^2 \lesssim \sqrt{\frac{\sigma^2/n_i + \sigma^3/\Delta}{T}} + \frac{\beta_i \sigma \Delta}{T^{\frac{1}{4}}(\sigma^2/n_i + \sigma^3/\Delta)^{\frac{1}{4}}}.$$

$\square$

**Lemma 13** (Variance reduction using Momentum). *Suppose that for all $i \in [N]$ and $x$,*

$$\mathbb{E}\|g_i(x) - \mathbb{E}_x g_i(x)\|^2 \le \sigma^2.$$

*Then*

$$\mathbb{E}\|m_{i,t} - \mathbb{E}_x m_{i,t}\|^2 \le \alpha \sigma^2.$$

*Proof of Lemma 13.*

$$\mathbb{E}\|m_{i,t} - \mathbb{E}m_{i,t}\|^2 = \mathbb{E}\|\alpha(g_i(x_{i,t-1}) - \nabla f_i(x_{i,t-1})) + (1 - \alpha)(m_{i,t-1} - \mathbb{E}m_{i,t-1})\|^2$$

$$\le \alpha^2 \mathbb{E}\|g_i(x_{i,t-1}) - \nabla f_i(x_{i,t-1})\|^2 + (1 - \alpha)^2 \mathbb{E}\|m_{i,t-1} - \mathbb{E}m_{i,t-1}\|^2$$

$$\le \alpha^2 \mathbb{E}\|g_i(x_{i,t-1}) - \nabla f_i(x_{i,t-1})\|^2 + (1 - \alpha)\mathbb{E}\|m_{i,t-1} - \mathbb{E}m_{i,t-1}\|^2$$

$$\le \alpha^2 \sigma^2 \sum_{q=0}^{t-1} (1 - \alpha)^q$$

$$\le \alpha^2 \sigma^2 \frac{(1 - \alpha)^t - 1}{(1 - \alpha) - 1}$$

$$\le \alpha^2 \sigma^2 \frac{1}{\alpha}$$

$$= \alpha \sigma^2.$$

$\square$

**Lemma 14** (Bound on Clustering Error).

$$\overline{\mathbb{E}\|v_{k_i^t, l_t} - \mathbb{E}_x \bar{m}_{i,t}\|^2} \lesssim \frac{\rho^2}{n_i} + \frac{\rho^3}{\Delta} + \beta_i \rho \Delta$$

*Proof of Lemma 14.* We prove the main result, and then justify each step afterwards.

$$\overline{\mathbb{E}\|v_{k_i,l_t,t} - \mathbb{E}_x \bar{m}_{i,t}\|^2} = \overline{\mathbb{E}\left\|\frac{1}{N}\sum_{j\in[N]} y_{j,l_t,t} - \mathbb{E}_x \bar{m}_{i,t}\right\|^2}$$

$$= \overline{\mathbb{E}\left\|(1-\beta_i)\left(\frac{1}{|\mathcal{G}_i|}\sum_{j\in\mathcal{G}_i} y_{j,l_t,t} - \mathbb{E}_x \bar{m}_{i,t}\right) + \beta_i\left(\frac{1}{|\mathcal{B}_i|}\sum_{j\in\mathcal{B}_i} y_{j,t,l} - \mathbb{E}_x \bar{m}_{i,t}\right)\right\|^2}$$

$$\overset{(i)}{\leq} (1+\beta_i)(1-\beta_i)^2 \overline{\mathbb{E}\left\|\left(\frac{1}{|\mathcal{G}_i|}\sum_{j\in\mathcal{G}_i} y_{j,l_t,t}\right) - \mathbb{E}_x \bar{m}_{i,t}\right\|^2}$$

$$+ \left(1+\frac{1}{\beta_i}\right)\beta_i^2 \overline{\mathbb{E}\left\|\left(\frac{1}{|\mathcal{B}_i|}\sum_{j\in\mathcal{B}_i} y_{j,l_t,t}\right) - \mathbb{E}_x \bar{m}_{i,t}\right\|^2}$$

$$\lesssim \overline{\mathbb{E}\left\|\left(\frac{1}{|\mathcal{G}_i|}\sum_{j\in\mathcal{G}_i} y_{j,l_t,t}\right) - \mathbb{E}_x \bar{m}_{i,t}\right\|^2} + \beta_i \overline{\mathbb{E}\left\|\left(\frac{1}{|\mathcal{B}_i|}\sum_{j\in\mathcal{B}_i} y_{j,l_t,t}\right) - \mathbb{E}_x \bar{m}_{i,t}\right\|^2}$$

$$\overset{(ii)}{\lesssim} (1-\delta_i+\beta_i)\overline{c_{k_i,l_t,t}^2} + \left(\frac{\rho^2}{n_i} + \frac{\rho^3}{\Delta} + \beta_i\rho\Delta\right)$$

$$\overset{(iii)}{\lesssim} (1-\delta_i/2)\overline{c_{k_i,l_t,t}^2} + \left(\frac{\rho^2}{n_i} + \frac{\rho^3}{\Delta} + \beta_i\rho\Delta\right)$$

$$\overset{(iv)}{\lesssim} \overline{(1-\delta_i/2)^{l_t} c_{k_i,1,t}^2} + \left(\frac{\rho^2}{n_i} + \frac{\rho^3}{\Delta} + \beta_i\rho\Delta\right)$$

$$\overset{(v)}{\leq} \frac{\rho}{\Delta}\overline{c_{k_i,1,t}^2} + \left(\frac{\rho^2}{n_i} + \frac{\rho^3}{\Delta} + \beta_i\rho\Delta\right). \tag{27}$$

We justify each step.

- (i) Young's inequality

- (ii) We prove this bound in Lemmas 15 and 19. Importantly, it shows that the clustering error is composed of two quantities: $\mathcal{E}_1$, the error contributed by good points from the cluster's perspective, and $\mathcal{E}_2$, the error contributed by the bad points from the cluster's perspective.

- (iii) Assumption that $\beta_i \lesssim \min(\delta_i, \delta_i/A^4)$

- (iv) Since $\mathbb{E}\|v_{i,l_t,t} - \mathbb{E}_x \bar{m}_{i,t}\|^2 = c_{k_i,l_t+1,t}^2$, the inequality forms a recursion which we unroll over $l_t$ steps.

- (v) Assumption that $l_t \geq \max(1, \frac{\log(\rho/\Delta)}{\log(1-\delta_i/2)})$

Finally, we note that

$$c_{k_i^t,1,t}^2 = \mathbb{E}\|\bar{m}_{i,t} - \mathbb{E}_x m_{i,t}\|^2$$
$$\lesssim \mathbb{E}\|m_{j,t} - \mathbb{E}_x m_{j,t}\|^2 + \mathbb{E}\|\mathbb{E}_x m_{j,t} - \mathbb{E}_x m_{i,t}\|^2$$
$$\leq \rho^2.$$

Applying this bound to (27) gives

$$\overline{\mathbb{E}\|v_{k_i,l_t,t} - \mathbb{E}_x \bar{m}_{i,t}\|^2} \lesssim \frac{\rho^2}{n_i} + \frac{\rho^3}{\Delta} + \beta_i\rho\Delta$$

$\square$

**Lemma 15** (Clustering Error due to Good Points)**.**

$$\overline{\mathbb{E}\left\|\left(\frac{1}{|\mathcal{G}_i|}\sum_{j\in\mathcal{G}_i} y_{j,l_t,t}\right) - \mathbb{E}_x \bar{m}_{i,t}\right\|^2} \lesssim (1-\delta_i)\overline{c_{k_i,l_t,t}^2} + \left(\frac{\rho^2}{n_i} + \frac{\rho^3}{\Delta}\right).$$

*Proof of Lemma 15.* We state the main sequence of steps and then justify them afterwards.

$$\mathbb{E}\left\|\left(\frac{1}{|\mathcal{G}_i|}\sum_{j\in\mathcal{G}_i} y_{j,l_t,t}\right) - \mathbb{E}_x \bar{m}_{i,t}\right\|^2$$

$$= \mathbb{E}\left\|\left(\frac{1}{|\mathcal{G}_i|}\sum_{j\in\mathcal{G}_i} y_{j,l_t,t}\right) - \frac{1}{|\mathcal{G}_i|}\sum_{j\in\mathcal{G}_i:j\sim i} \mathbb{E}_x m_{j,t} - \left(\frac{1}{n_i} - \frac{1}{|\mathcal{G}_i|}\right)\sum_{j\in\mathcal{G}_i:j\sim i} \mathbb{E}_x m_{j,t}\right\|^2$$

$$= \mathbb{E}\left\|\left(\frac{1}{|\mathcal{G}_i|}\sum_{j\in\mathcal{G}_i:j\sim i} (y_{j,l_t,t} - \mathbb{E}_x m_{j,t})\right) + \left(\frac{1}{|\mathcal{G}_i|}\sum_{j\in\mathcal{G}_i:j\not\sim i} (y_{j,l_t,t} - \mathbb{E}_x \bar{m}_{i,t})\right)\right\|^2$$

$$\overset{(i)}{\leq} \left(1+\frac{2}{\delta_i}\right)\mathbb{E}\left\|\frac{1}{|\mathcal{G}_i|}\sum_{j\in\mathcal{G}_i:j\sim i} (y_{j,l_t,t} - \mathbb{E}_x m_{j,t})\right\|^2 + \left(1+\frac{\delta_i}{2}\right)\mathbb{E}\left\|\frac{1}{|\mathcal{G}_i|}\sum_{j\in\mathcal{G}_i:j\not\sim i} (y_{j,l_t,t} - \mathbb{E}_x \bar{m}_{i,t})\right\|^2$$

$$\overset{(ii)}{\lesssim} \left(1+\frac{2}{\delta_i}\right)\mathbb{E}\left\|\frac{1}{|\mathcal{G}_i|}\sum_{j\in\mathcal{G}_i:j\sim i} \mathbb{E}y_{j,l_t,t} - \mathbb{E}_x m_{j,t}\right\|^2 + \left(1+\frac{2}{\delta_i}\right)\mathbb{E}\left\|\frac{1}{|\mathcal{G}_i|}\sum_{j\in\mathcal{G}_i:j\sim i} (y_{j,l_t,t} - \mathbb{E}_x y_{j,l_t,t})\right\|^2$$

$$+ \left(1+\frac{\delta_i}{2}\right)\mathbb{E}\left\|\frac{1}{|\mathcal{G}_i|}\sum_{j\in\mathcal{G}_i:j\not\sim i} (y_{j,l_t,t} - \mathbb{E}_x \bar{m}_{i,t})\right\|^2$$

$$\overset{(iii)}{\lesssim} \left(1+\frac{2}{\delta_i}\right)\mathbb{E}\left\|\frac{1}{|\mathcal{G}_i|}\sum_{j\in\mathcal{G}_i:j\sim i} \mathbb{E}(y_{j,l_t,t} - m_{j,t})\right\|^2 + \left(1+\frac{2}{\delta_i}\right)\mathbb{E}\left\|\frac{1}{|\mathcal{G}_i|}\sum_{j\in\mathcal{G}_i:j\sim i} (\mathbb{E}m_{j,t} - \mathbb{E}_x m_{j,t})\right\|^2$$

$$+ \left(1+\frac{2}{\delta_i}\right)\mathbb{E}\left\|\frac{1}{|\mathcal{G}_i|}\sum_{j\in\mathcal{G}_i:j\sim i} (y_{j,l_t,t} - \mathbb{E}y_{j,l_t,t})\right\|^2 + \left(1+\frac{\delta_i}{2}\right)\mathbb{E}\left\|\frac{1}{|\mathcal{G}_i|}\sum_{j\in\mathcal{G}_i:j\not\sim i} (y_{j,l_t,t} - \mathbb{E}_x \bar{m}_{i,t})\right\|^2$$

$$\overset{(iv)}{\lesssim} \left(1+\frac{2}{\delta_i}\right)\delta_i^2 \underbrace{\mathbb{E}_{j\in\mathcal{G}_i:j\sim i}\|\mathbb{E}_x(y_{j,l_t,t} - m_{j,t})\|^2}_{\mathcal{T}_1} + \left(1+\frac{2}{\delta_i}\right)\frac{n_i}{|\mathcal{G}_i|^2}\rho^2$$

$$+ \left(1+\frac{2}{\delta_i}\right)\frac{n_i}{|\mathcal{G}_i|^2}\underbrace{\mathbb{E}\left\|\frac{1}{|\mathcal{G}_i|}\sum_{j\in\mathcal{G}_i:j\sim i} (y_{j,l_t,t} - \mathbb{E}y_{j,l_t,t})\right\|^2}_{\mathcal{T}_2} + \left(1+\frac{\delta_i}{2}\right)(1-\delta_i)^2 \underbrace{\mathbb{E}_{j\in\mathcal{G}_i:j\not\sim i}\|y_{j,l_t,t} - \mathbb{E}_x \bar{m}_{i,t}\|^2}_{\mathcal{T}_3}$$

$$\overset{(v)}{\lesssim} \delta_i \mathbb{E}_{j\in\mathcal{G}_i:j\sim i}\|\mathbb{E}_x(y_{j,l_t,t} - m_{j,t})\|^2 + \frac{\rho^2}{n_i} + \left(1+\frac{\delta_i}{2}\right)(1-\delta_i)^2 \mathbb{E}_{j\in\mathcal{G}_i:j\not\sim i}\|y_{j,l_t,t} - \mathbb{E}_x \bar{m}_{i,t}\|^2$$

$$\overset{(vi)}{\lesssim} \delta_i\left(\overline{c_{k_i,l_t,t}^2} + \frac{\rho^3}{\delta_i\Delta}\right) + \frac{\rho^2}{n_i} + \left(1+\frac{\delta_i}{2}\right)(1-\delta_i)^2\left(\left(1+\frac{\delta_i}{2}\right)\overline{c_{k_i,l_t,t}^2} + \frac{\rho^3}{\Delta}\right)$$

$$\lesssim \delta_i\left(\overline{c_{k_i,l_t,t}^2} + \frac{\rho^3}{\delta_i\Delta}\right) + \frac{\rho^2}{n_i} + (1-\delta_i)\left(\overline{c_{k_i,l_t,t}^2} + \frac{\rho^3}{\Delta}\right)$$

$$\lesssim (1-\delta_i)\overline{c_{k_i,l_t,t}^2} + \left(\frac{\rho^2}{n_i} + \frac{\rho^3}{\Delta}\right).$$

Justifications for the labeled steps are:

- (i), (ii), (iii) Young's inequality

- (iv) First, we can can interchange the sum and the norm due to independent stochasticity of the momentums. Then, by the Tower Property and Law of Total Variance for the 1st and 3rd steps

respectively

$$
\begin{aligned}
\mathbb{E}\|\mathbb{E}_x m_{j,t} - \mathbb{E}m_{j,t}\|^2 &= \mathbb{E}\|\mathbb{E}_x m_{j,t} - \mathbb{E}[\mathbb{E}_x m_{j,t}]\|^2 \\
&= \mathrm{Var}(\mathbb{E}_x m_{j,t}) \\
&= \mathrm{Var}(m_{j,t}) - \mathbb{E}(\mathrm{Var}_x(m_{j,t})) \\
&= \mathrm{Var}(m_{j,t}) - \mathbb{E}\|m_{j,t} - \mathbb{E}_x m_{j,t}\|^2 \\
&\lesssim \rho^2,
\end{aligned}
$$

where the last inequality follows since the two terms above it are both bounded by $\rho^2$.

- (v) We prove this bound in Lemmas 16, 17, and 18. It shows that, from point $i$'s perspective, the error of its cluster-center-estimate is composed of three quantities: $\mathcal{T}_1$, the error introduced by our thresholding procedure on the good points which belong to $i$'s cluster (and therefore ideally are included within the thresholding radius); $\mathcal{T}_2$, which accounts for the variance of the clipped points in $i$'s cluster; and $\mathcal{T}_3$, the error due to the good points which don't belong to $i$'s cluster (and therefore ideally are forced outside the thresholding radius).

- (vi) $(1 + {}^x\!/_2)^2(1-x)^2 \le 1 - x$ for all $x \in [0,1]$

$\qquad\qquad\qquad\qquad\qquad\qquad\qquad\qquad\qquad\qquad\qquad\qquad\qquad\qquad\qquad\qquad\qquad$ □

**Lemma 16** (Bound $\mathcal{T}_1$: Error due to In-Cluster Good Points)**.**

$$
\overline{\mathbb{E}_{j \in \mathcal{G}_i : j \sim i}\|\mathbb{E}(y_{j,l_t,t} - m_{j,t})\|^2} \lesssim \overline{c^2_{k_i,l_t,t}} + \frac{\rho^3}{\delta_i \Delta}.
$$

*Proof of Lemma 16.* By definition of $y_{j,l_t,t}$,

$$
\begin{aligned}
\mathbb{E}\|y_{j,l_t,t} - m_{j,t}\| &= \mathbb{E}[\|v_{k_i,l_t-1,t} - m_{j,t}\|\mathbb{1}(\|v_{k_i,l_t-1,t} - m_{j,t}\| > \tau_{k_i,l_t,t})] \\
&\le \frac{\mathbb{E}[\|v_{k_i,l_t-1,t} - m_{j,t}\|^2 \mathbb{1}(\|v_{k_i,l_t-1,t} - m_{j,t}\| > \tau_{k_i,l_t,t})]}{\tau_{k_i,l_t,t}}.
\end{aligned}
$$

Therefore by Jensen's inequality,

$$
\begin{aligned}
\|\mathbb{E}y_{j,l_t,t} - m_{j,t}\|^2 &\le (\mathbb{E}\|y_{j,l_t,t} - m_{j,t}\|)^2 \\
&\le \frac{(\mathbb{E}\|v_{k_i,l_t-1,t} - m_{j,t}\|^2)^2}{\tau^2_{k_i,l_t,t}} \\
&\lesssim \frac{(\mathbb{E}\|v_{k_i,l_t-1,t} - \mathbb{E}_x \bar{m}_{i,t}\|^2 + \mathbb{E}\|\mathbb{E}_x \bar{m}_{i,t} - m_{j,t}\|^2)^2}{\tau^2_{k_i,l_t,t}} \\
&= \frac{(\mathbb{E}\|v_{k_i,l_t-1,t} - \mathbb{E}_x \bar{m}_{i,t}\|^2 + \mathbb{E}\|\mathbb{E}_x m_{j,t} - m_{j,t}\|^2)^2}{\tau^2_{k_i,l_t,t}} \\
&\le \frac{(c^2_{k_i,l_t,t} + \rho^2)^2}{\tau^2_{k_i,l_t,t}} \\
&\lesssim \frac{(c^2_{k_i,l_t,t} + \rho^2)^2}{c^2_{k_i,l_t,t} + \delta_i \rho \Delta} \\
&\lesssim \left(1 + \frac{\rho}{\delta_i \Delta}\right)c^2_{k_i,l_t,t} + \frac{\rho^3}{\delta_i \Delta} \\
&\lesssim c^2_{k_i,l_t,t} + \frac{\rho^3}{\delta_i \Delta}.
\end{aligned}
$$

Summing this inequality over $t$ and dividing by $T$, we have

$$\overline{\mathbb{E}_{j \in \mathcal{G}_i : j \sim i} \| \mathbb{E}_x (y_{j,l_t,t} - m_{j,t}) \|^2} \leq \overline{\mathbb{E} \| y_{j,l_t,t} - m_{j,t} \|^2}$$
$$\lesssim \overline{c_{k_i,l_t,t}^2} + \frac{\rho^3}{\delta_i \Delta}.$$

$\square$

**Lemma 17** (Bound $\mathcal{T}_2$: Variance of Clipped Points).

$$\mathbb{E} \left\| \frac{1}{|\mathcal{G}_i|} \sum_{j \in \mathcal{G}_i : j \sim i} (y_{j,l_t,t} - \mathbb{E} y_{j,l_t,t}) \right\|^2 \leq \frac{n_i}{|\mathcal{G}_i|^2} \rho^2.$$

*Proof of Lemma 17.* Note that the elements in the sum $\sum_{j \in \mathcal{G}_i : j \sim i} (y_{j,l_t,t} - \mathbb{E} y_{j,l_t,t})$ are not independent. Therefore, we cannot get rid of the cross terms when expanding the squared-norm. However, if for each round of thresholding we sampled a fresh batch of points to set the new cluster-center estimate, then the terms would be independent. With this resampling strategy, our bounds would only change by a constant factor. Therefore, for ease of analysis, we will assume the terms in the sum are independent. In that case,

$$\mathbb{E} \left\| \frac{1}{|\mathcal{G}_i|} \sum_{j \in \mathcal{G}_i : j \sim i} (y_{j,l_t,t} - \mathbb{E} y_{j,l_t,t}) \right\|^2 \leq \frac{n_i}{|\mathcal{G}_i|^2} \mathbb{E} \| y_{j,l_t,t} - \mathbb{E} y_{j,l_t,t} \|^2$$
$$\leq \frac{n_i}{|\mathcal{G}_i|^2} \mathbb{E} \| m_{j,t} - \mathbb{E} m_{j,t} \|^2$$
$$\leq \frac{n_i}{|\mathcal{G}_i|^2} \rho^2,$$

where the second-to-last inequality follows from the contractivity of the thresholding procedure. $\square$

**Lemma 18** (Bound $\mathcal{T}_3$: Error due to Out-of-Cluster Good Points).

$$\overline{\mathbb{E}_{j \in \mathcal{G}_i : j \not\sim i} \| y_{j,l_t,t} - \mathbb{E}_x \bar{m}_{i,t} \|^2} \lesssim \left( 1 + \frac{\delta_i}{2} \right) \overline{c_{k_i,l_t,t}^2} + \frac{\rho^3}{\Delta}.$$

*Proof of Lemma 18.* This sequence of steps bounds the clustering error due to points not from client $i$'s cluster. Using Young's inequality for the first step,

$$\mathbb{E}_{j \in \mathcal{G}_i : j \not\sim i} \| y_{j,l_t,t} - \mathbb{E}_x \bar{m}_{i,t} \|^2$$
$$\leq \left( 1 + \frac{\delta_i}{2} \right) \mathbb{E} \| v_{k_i^t, l_t - 1} - \mathbb{E}_x \bar{m}_{i,t} \|^2 + \left( 1 + \frac{2}{\delta_i} \right) \mathbb{E}_{j \in \mathcal{G}_i : j \not\sim i} \| y_{j,l_t,t} - v_{k_i, l_t - 1, t} \|^2$$
$$\leq \left( 1 + \frac{\delta_i}{2} \right) c_{k_i,l_t,t}^2 + \left( 1 + \frac{2}{\delta_i} \right) \mathbb{E}_{j \in \mathcal{G}_i : j \not\sim i} \| y_{j,l_t,t} - v_{k_i, l_t - 1, t} \|^2$$
$$= \left( 1 + \frac{\delta_i}{2} \right) c_{k_i,l_t,t}^2 + \left( 1 + \frac{2}{\delta_i} \right) \mathbb{E}_{j \in \mathcal{G}_i : j \not\sim i} [ \| m_{j,t} - v_{k_i, l_t - 1, t} \|^2 \mathbb{1} \{ \| m_{j,t} - v_{k_i, l_t - 1, t} \| \leq \tau_{k_i,l_t,t} \} ]$$
$$\leq \left( 1 + \frac{\delta_i}{2} \right) c_{k_i,l_t,t}^2 + \left( 1 + \frac{2}{\delta_i} \right) \tau_{k_i,l_t,t}^2 \mathbb{P}_{j \in \mathcal{G}_i : j \not\sim i} ( \| m_{j,t} - v_{k_i, l_t - 1, t} \| \leq \tau_{k_i,l_t,t} )$$
$$\lesssim \left( 1 + \frac{\delta_i}{2} \right) c_{k_i,l_t,t}^2 + \left( \frac{1}{\delta_i} \right) (c_{k_i,l_t,t}^2 + \delta_i \rho \Delta) \mathbb{P}_{j \in \mathcal{G}_i : j \not\sim i} ( \| m_{j,t} - v_{k_i^t, l_t - 1} \| \leq \tau_{k_i,l_t,t} ).$$

If $\| v_{k_i^t, l_t - 1} - m_{j,t} \| \leq \tau_{k_i,l_t,t}$, then

$$\| \mathbb{E}_x m_{j,t} - \mathbb{E}_x m_{i,t} \|^2 \lesssim \| m_{j,t} - \mathbb{E}_x m_{j,t} \|^2 + \| m_{j,t} - v_{k_i, l_t - 1, t} \|^2 + \| v_{k_i^t, l_t - 1} - \mathbb{E}_x \bar{m}_{i,t} \|^2$$
$$\lesssim \| m_{j,t} - \mathbb{E}_x m_{j,t} \|^2 + \tau_{k_i,l_t,t}^2 + \| v_{k_i, l_t - 1, t} - \mathbb{E}_x \bar{m}_{i,t} \|^2$$

By Assumption 9,

$$\Delta^2 \lesssim \|m_{j,t} - \mathbb{E}_x m_{j,t}\|^2 + \tau^2_{k_i,l_t,t} + \|v_{k_i,l_t-1,t} - \mathbb{E}_x \bar{m}_{i,t}\|^2$$

which, summing over $t$ and dividing by $T$, implies

$$\overline{\|m_{j,t} - \mathbb{E}_x m_{j,t}\|^2} + \overline{\|v_{k_i,l_t-1,t} - \mathbb{E}_x \bar{m}_{i,t}\|^2} \gtrsim \Delta^2 - \overline{\tau^2_{k_i,l_t,t}}.$$

By Markov's inequality, the probability of this event is upper-bounded by

$$\frac{\rho^2 + \overline{\mathbb{E}\|v_{k_i,l_t-1,t} - \mathbb{E}_x \bar{m}_{i,t}\|^2}}{\Delta^2 - \overline{\tau^2_{k_i,l_t,t}}} \lesssim \frac{\rho^2 + \overline{c^2_{k_i,l_t,t}}}{\Delta^2},$$

where the inequality holds due to the constraint on $\Delta$ from the theorem statement. Therefore,

$$\overline{\mathbb{E}_{j \in \mathcal{G}_i : j \not\sim i} \|y_{j,l_t,t} - \mathbb{E}_x \bar{m}_{i,t}\|^2} \lesssim \left(1 + \frac{\delta_i}{2} + \frac{\rho}{\Delta} + \frac{\rho^2 + \overline{c^2_{k_i,l_t,t}}}{\delta_i \Delta^2}\right) \overline{c^2_{k_i,l_t,t}} + \frac{\rho^3}{\Delta}$$

$$\lesssim \left(1 + \frac{\delta_i}{2}\right) \overline{c^2_{k_i,l_t,t}} + \frac{\rho^3}{\Delta},$$

where again we apply the constraint on $\Delta$ for the second inequality. $\square$

**Lemma 19** (Clustering Error due to Bad Points).

$$\overline{\mathbb{E}\left\|\left(\frac{1}{|\mathcal{B}_i|} \sum_{j \in \mathcal{B}_i} y_{j,l_t,t}\right) - \mathbb{E}_x \bar{m}_{i,t}\right\|^2} \lesssim \overline{c^2_{k_i,l_t,t}} + \delta_i \rho \Delta$$

*Proof of Lemma 19.* This lemma gives a bound on the clustering error due to the bad clients from client $i$'s perspective. The goal of such clients would be to corrupt the cluster-center estimate of client $i$'s cluster as much as possible at each round. They can have the maximum negative effect by setting their gradients to be just inside the thresholding radius around client $i$'s cluster-center estimate. This way, the gradients will keep their value (rather than be assigned the value of the current cluster-center estimate per our update rule), but they will have maximal effect in moving the cluster-center estimate from its current position. Therefore, in step 3 of the inequalities below, we can not do better than bounding the distance between these bad points and the current cluster center estimate (i.e. $\|y_{j,l_t,t} - v_{k_i,l_t-1,t}\|^2$) by the thresholding radius $\tau_{k_i,l_t,t}$.

$$\mathbb{E}\left\|\left(\frac{1}{|\mathcal{B}_i|} \sum_{j \in \mathcal{B}_i} y_{j,l_t,t}\right) - \mathbb{E}_x \bar{m}_{i,t}\right\|^2 \leq \mathbb{E}_{j \in \mathcal{B}_i} \|y_{j,l_t,t} - \mathbb{E}_x \bar{m}_{i,t}\|^2$$

$$\lesssim \mathbb{E}_{j \in \mathcal{B}_i} \|y_{j,l_t,t} - v_{k_i,l_t-1,t}\|^2 + \mathbb{E}\|v_{k_i,l_t-1,t} - \mathbb{E}_x \bar{m}_{i,t}\|^2$$

$$\leq \tau^2_{k_i,l_t,t} + c^2_{k_i,l_t,t}$$

$$\lesssim c^2_{k_i,l_t,t} + \delta_i \rho \Delta.$$

The last inequality applies the definition of $\tau_{k_i,l_t,t}$, and the result of the lemma follows by summing this inequality over $t$ and dividing by $T$. $\square$

