# OpenReview forum: "Provably Personalized and Robust Federated Learning"
_TMLR — Accepted by TMLR_

### Review · Reviewer_Abdu · 2023-08-13

**Summary Of Contributions:**

This work studies the problem in the intersection of personalization and Byzantine robustness. The authors show the failure of previous clustering-type methods on simple constructed examples and suggest a new algorithm: Federated-Clustering (FC).
The proposed approach is equipped with rigorous "clustering" and robustness guarantees. In addition, the paper extends results to non-convex smooth stochastic optimization under assumptions of Intra-cluster Similarity and Inter-cluster Separation of the gradients. Communication efficiency is improved  by using momentum under stronger assumptions on momentum parameters rather than gradients.
Simple experiments validate some of the theoretical assumptions and show the benefits of FC on a synthetic quadratic problem and MNIST against other personalization algorithms.

**Audience:**

Yes

**Broader Impact Concerns:**

I do not foresee negative ethical implications of the work due to its mainly theoretical nature.

**Claims And Evidence:**

Yes

**Requested Changes:**

I believe this work can be accepted in its current form. However, next, I list some suggestions which, in my view, may improve and strengthen the work.

The paper needs a more detailed discussion on compatibility with modern Private Federated Learning systems. An explicit limitations section of the obtained theoretical results would also be very beneficial.

The experiments description is relatively short and does not explain how and why the hyperparameters were chosen (e.g., for Ditto). I am afraid the results are not reproducible due to a lack of details.

It would be helpful to discuss how big the lower bound for $l$ (from Theorem 1) can be, maybe by plugging some expected values or through experimental evaluation.

I would like to ask the authors to add a more detailed comparison to prior works' theoretical results, i.e., Ghosh et al. (2021). In addition, I suggest putting this work into a broader context of Byzantine-robust optimization. For instance, by adding a comparison to the work [R] and how obtained results build on previos papers, what are the innovations in the analysis, and so on.

It would be ideal to have a more realistic federated experimental setting, e.g., on a subset of Federated EMNIST.

It is encouraged to make the proofs in the Appendix more comprehensible by splitting the long sequences of inequalities.

There are some minor notation issues.

1. The $\stackrel{i. i. d.}{\sim}$ sign meaning (in Assumption 1) is not fully clear as this particular notation is not defined. Does it mean that points $z_i, z_j$ are random vectors with the same probability distribution?

2. The main text does not introduce the Batch size $|B|$ (in Theorem 3). Algorithm 2 (FC) does not mention any mini-batching. It seems that the lower bound for $|B|$ may be larger than the size $m_i$ of the client $i$'s training dataset, which may be problematic.

3. Does generating matrices $A_i$ from _"gaussian distribution"_ refer to sampling its entries and $y_i$ from __standard__ normal?

Typo in the reference
> H. Brendan **McMahon**, Eider Moore, Daniel Ramage, Seth Hampson, and Blaise Aguera y Arcas. Communication-efficient learning of deep networks from decentralized data. In Proceedings of the 20th Conference on Artificial Intelligence and Statistics, volume 54, 2017

___
[R] Karimireddy, Sai Praneeth, Lie He, and Martin Jaggi. "Learning from history for byzantine robust optimization." International Conference on Machine Learning. PMLR, 2021.

**Strengths And Weaknesses:**

# Strengths

This work is the first approach to combining personalization with Byzantine robustness in a quite general distributed learning setting. In my view, this is a very **exciting research direction** as it allows to provably extend distributed robustness results to heterogeneous settings (essential for Federated Learning), which was hard to do before.

**The writing** is clear, and the presentation makes the paper easy to follow. It has an understandable and well-organized structure, starting from simple examples, then introducing the Threshold-Clustering subroutine, and gradually extending the results to more general and practical settings.

**Theoretical analysis** is based on quite reasonable assumptions. The estimation error of the suggested clustering method is supported with a simple lower bound. Helpful explanatory comments accompany theorems and assumptions. It is commendable that the authors also extend the proposed Federated-Clustering to a more practical momentum-based method.

In addition, simple **experiments** serve as a good illustrative demonstration of the interesting theoretical results.

# Weaknesses

One of the proposed algorithms: Federated-Clustering, can be very problematic in **private** federated learning as it requires broadcasting personalized models to other clients, which may be infeasible in practical systems.

Moreover, theoretical guarantees for FC require setting thresholding radius $\tau$, which depends on (in general) unknown quantities. A similar problem holds for the step size upper bound in Theorem 4, which depends, i.e., on $f_i^*$.

Assumptions 4 and 5 may be problematic from an optimization perspective due to a potential restriction of the class of functions. Empirical estimation on a synthetic non-adversarial problem is not very convincing.

---

> ### Author Response · Authors · 2023-08-28
> **Response to Reviewer Abdu**
>
> Thank you for taking the time to read our paper thoroughly and for providing such detailed and helpful feedback. We have incorporated all requested changes. We address each of the **Requested Changes** and **Weaknesses** raised.
>
> **Requested Changes**
>
> 1. **Discussion on privacy in FL...**: We acknowledge that our algorithm only has a light layer of privacy, sharing models and gradients instead of raw data. Since the focus of our work is not on privacy, but rather on achieving near-optimal convergence guarantees, we defer a more rigorous treatment of privacy to future work. We have added a section after the analysis of our algorithm that contextualizes our work in the larger FL-privacy literature.
>
> 2. **Include more experiments...**: In Section 4.5, we have added simulations to verify that hyperparameter $l$ in Theorem 1 scales with $\log{\sigma}$ as stated. We are also running additional experiments on CIFAR10 and CIFAR100, and will update as soon as we have the results. In the revision, we have also expanded details on existing experiments -- choice of hyperparameters, motivation for experiment design, descriptions for captions. In the case of Ditto, we grid searched the hyperparameter $\lambda_{ditto}$ and found that the best choice is $\lambda_{ditto}=0$ which degenerates "Local training". Therefore, we pick $\lambda_{ditto}=1$ to demonstrate non-trivial results. All code for experiments is in the .zip file submitted as supplementary material.
>
> 3. **Lower bound for $l$...**: We have provided additional insights on this in Section 4.5. We executed a series of clustering tasks with a consistent $\Delta$ and varying $\sigma$. Our observations reveal that the minimum $l$ required for convergence increases in tandem with $\sigma$, and this relationship aligns quantitatively with the $\log{\sigma}$ scaling stated in Theorem 1.
>
> 4. **Comparison to prior works...**: We have expanded the related work section to incorporate these suggestions. We have added a detailed comparison between our work and Ghosh et al. (2021), discussed how our work relates to the existing literature on Byzantine-robust optimization along with a comparison to Karimireddy et al. (2021), and provided additional discussion of two recent papers ([Wu] and [Wang]) on personalized federated learning.
>
> 5. **More realistic experiments...**: We are currently running new experiments on CIFAR10 and CIFAR100 for the final revision in addition to our existing experiments on MNIST, and will update as soon as we have the results.
>
> 6. **Readability of proofs...**: Thank you for the feedback. We already spent considerable effort in making the proof presentable. Unfortunately, it has many moving parts and is tricky. We will carefully go over it again and see if there any further improvements we can make.
>
> 7. **Notation**
>
>       a. **i.i.d. notation...**: Yes your interpretation is how we meant it. We have made this explicit in revision.
>
>       b. **Batch size...**: Thank you for this suggestion. We now introduce and discuss batch size in the main text, have edited the bound on $|B|$ in Theorem 3 to account for the case $|B|=m_i$, and discuss what happens when $|B|=m_i$ after the theorem statement.  In particular, when $|B|=m_i$, the variance of the stochastic gradients is $0$, and we recover the $\mathcal{O}(1/T)$ rate for deterministic gradient descent (see eq. (16) on pg. 22 in proof). Theorem 3 is meant to highlight the effect of parameters $\sigma$, $n_i$ and $\Delta$ on convergence rate, so we focus on the $|B|<m_i$ case in the theorem but give full detail in the proof. We also note that as long as there are no malicious clients (i.e. $\beta_i=0$), there are a large enough number of clients $n_i$ in the cluster, and inter-cluster separation $\Delta$ is sufficiently larger than the intra-cluster variance $\sigma^2$, then minimum batch size will likely be less than $m_i$. Finally, we note that the batch-size constraint for Federated-Clustering reduces the variance of stochastic gradients. In Section 3.3, we propose an alternative algorithm Momentum-Clustering with no restriction on batch-size that reduces variance by clustering momentums instead of gradients.
>
>      c. **Distribution of sample matrix...**: For client $i$ in cluster $k$, we draw data matrix $A_i\overset{i.i.d.}{\sim}\mathcal{N}(k, \mathbf{1}_{d\times n})$ where $d$ is the number of features and $n$ is the number of local samples. In this way, clients in the same cluster have same data distribution while clients from different clusters are well separated. The $y_i$ is deterministically computed through $y_i=A_i^\top\mathbf{x}_k^\star$. In the experiment, we let $d>n$ and $d<nn_i$ ($n_i$ is the number of clients in the same cluster as client $i$) so that a client alone cannot uniquely determine $\mathbf{x}^\star$ but clients in the same cluster can uniquely find $\mathbf{x}^\star$.
>
>      d. **References typos**: Thank you for catching this. We have fixed the typos.

---

> > ### Author Response · Authors · 2023-08-28
> > **Response to Reviewer Abdu cont.**
> >
> > **Weaknesses**
> >
> > 1. **Unknown hyperparameters...**: While these quantities are unknown, in practice we can use heuristics to approximate them. For instance, in our MNIST experiment, we set $\tau$ to be the 20th-percentile of distances of client gradients to the cluster-center, and we use a grid-search to set the step-size. Developing novel methods free of such hyper-parameters is an excellent question for future work.
> >
> > 2. **Justification for Assumptions 4 and 5...**: We have added a few paragraphs to the revision justifying these assumptions (directly after stating them). When clients within a cluster are *i.i.d.*, Assumption 4 is trivially satisfied for all functions (the left-side of the inequality in Assumption 4 becomes $0$). When clusters are well separated -- meaning at any given parameter $x$, the loss functions of clients from different clusters are far apart when evaluated at $x$ -- Assumption 5 is satisfied. Therefore, with no restrictions on the function classes, Assumptions 4 and 5 can be easily satisfied when clients are *i.i.d.* within a cluster and clusters are well-separated. Notably however, Assumptions 4 and 5 permit significantly more complex settings. Namely, with Assumption 4, we do not require that clients within a cluster are *i.i.d.*, only that the distance between their gradients and the cluster-average gradient scales does not exceed the scaled size of the cluster-average gradient. With Assumption 5, we do not require that clients from different clusters are well-separated at *all* parameters values, only at their respective optima.
> >
> > *References*
> >
> > [Wu] Wu, Yue and Zhang, Shuaicheng and Yu, Wenchao and Liu, Yanchi and Gu, Quanquan and Zhou, Dawei and Chen, Haifeng and Cheng, Wei. "Personalized federated learning under mixture of distributions". Proceedings of the 40th International Conference on Machine Learning (PMLR), 2023.
> >
> > [Wang] Wang, Tianchun and Cheng, Wei and Luo, Dongsheng and Yu, Wenchao and Ni, Jingchao and Tong, Liang and Chen, Haifeng and Zhang, Xiang. "Personalized federated learning via heterogeneous modular networks". 2022.

---

### Review · Reviewer_C9QW · 2023-08-18

**Summary Of Contributions:**

The paper proposes a novel framework for personalized federated learning that clusters clients with similar objectives and learns a model per cluster. The authors provide optimal convergence guarantees and provable robustness in the Byzantine setting. The proposed algorithm is evaluated on synthetic and MNIST datasets, and the results show that it outperforms existing federated learning algorithms in terms of both accuracy and communication efficiency.

Overall, the paper presents a significant contribution to the field of federated learning by addressing the challenge of personalization and robustness in a provable manner. The proposed algorithm has the potential to be applied in various real-world scenarios where personalization and robustness are crucial, such as healthcare and finance. The paper provided thorough proofs. However, the experiments are not sufficient. Only one real datasets are used. Also, some important references are missing.




**Audience:**

Yes

**Claims And Evidence:**

Yes

**Requested Changes:**

- The experimental need to include more real datasets, such as CIFAIR10, CIFAIR100, etc.
- Detailed analysis of the communication overhead of the proposed algorithm is needed.
-  Provide a comprehensive analysis of related works and highlights the differences and similarities between the proposed algorithm and existing approaches. Compare with more recent SOTA approaches, such as the listed above.

**Strengths And Weaknesses:**

Pros:
- The paper addresses the challenge of personalization in federated learning, which is a crucial problem in many real-world scenarios.
- The proposed algorithm clusters clients with similar objectives and learns a model per cluster, which is an intuitive and interpretable approach to personalization.
- The authors provide optimal convergence guarantees and provable robustness in the Byzantine setting, which is a significant contribution to the field of federated learning.
- The proposed algorithm outperforms existing federated learning algorithms in terms of both accuracy and communication efficiency, as shown in the experimental evaluation.


Cons:
- The experimental evaluation is limited to a few datasets, nly one real datasets are used. and it's unclear how the proposed algorithm would perform on larger and more diverse datasets.
- The paper does not provide a detailed analysis of the computational overhead of the proposed algorithm, which may be a significant factor in some scenarios.
- The paper did not provide a comprehensive analysis of related works and highlights the differences and similarities between the proposed algorithm and existing approaches. Some important references are missing such as:
Personalized Federated Learning under Mixture of Distributions.
International Conference on Machine Learning (ICML'23), 2023.
Personalized Federated Learning via Heterogeneous Modular Networks.
In 2022 IEEE International Conference on Data Mining (ICDM'22).

---

> ### Author Response · Authors · 2023-08-28
> **Response to Reviewer C9QW**
>
> Thank you for your very valuable feedback, all of which we incorporated to improve the paper. We address each of your **Requested Changes** here.
>
> **Requested Changes**
>
> 1. We are currently running experiments on CIFAR10 and CIFAR100 and will update the revision with these as soon as we have the results. Per Reviewer Abdu's suggestion, we have also added simulations in Section 4.5 to verify that hyperparameter $l$ in Theorem 1 scales with $\log{\sigma}$ as stated.
>
> 2. We have added a section (directly before Section 3.3) discussing the communication overhead of our algorithms Federated-Clustering and Momentum-Clustering.
>
> 3. We have expanded our related work section to incorporate all of these suggestions. We now give a detailed comparison of our work and the relevant existing work on clustering methods for personalized federated learning. We also discuss the two papers you reference ([Wu] and [Wang]), and clarify how our work relates to existing literature on Byzantine-robust optimization.
>
> *References*
>
> [Wu] Wu, Yue and Zhang, Shuaicheng and Yu, Wenchao and Liu, Yanchi and Gu, Quanquan and Zhou, Dawei and Chen, Haifeng and Cheng,  Wei. "Personalized federated learning under mixture of distributions". Proceedings of the 40th International Conference on Machine Learning (PMLR), 2023.
>
> [Wang] Wang, Tianchun and Cheng, Wei and Luo, Dongsheng and Yu, Wenchao and Ni, Jingchao and Tong, Liang and Chen, Haifeng and Zhang, Xiang. "Personalized federated learning via heterogeneous modular networks". 2022.

---

### Review · Reviewer_SJcN · 2023-08-21

**Summary Of Contributions:**

The paper proposes a robust clustering framework on the personalized federated learning setting. Specifically, it first analyzes several failure modes that the current framework experienced and then propose the threshold clustering method. It keeps the data points unchanged under a set certain radius and scales those that are outside the radius. The theoretical analysis have been first conducted under iid data distribution and then extended into client's gradient in the framework. To speed up the procedure, it later proposes to use client's momentum. The experiments have been done on a synthetic data and MNIST dataset to show the proposed method could achieve comparable and even better performance than baselines.

**Audience:**

Yes

**Broader Impact Concerns:**

No need for ethical review.

**Claims And Evidence:**

No

**Requested Changes:**

1. I suggest to move the failure modes/cases into the motivation before the proposed method.
2. Some description on the Algorithm 1 is needed. Personally I don't think it is a good idea to just introduce the algorithm by only giving the algorithm itself.
3. Polish Figure 1 and 2. Add description on axises and make the text bigger and visible. Also, consider to optimize the space for the Figure. For example, just put all three figures in Figure 1 in a single row.
4. Include some malicious users cases in the simulation.

**Strengths And Weaknesses:**

Pros:
1. The paper do a lot of theoretical analysis and proposes a new clustering algorithm that could be used into personalized federated learning setting.

Cons:
1. The paper needs major revision on the paper organization. The algorithm is introduced very late however the comparison and discussion have been already discussed, which makes the paper difficult to follow.
2. The paper presentation needs to be polished. Figure 1 is to small to be viewed. Figure 1 and 2 don't have explanation on the axises. I find myself very hard to understand those results and why the Figure 1(c) is outperformed than (a) and (b).
3. The examples in page 2,3,4 need further explanation. I don't get it why the gradient on the first page should be $1/2eta$, $1/2eta$, $-1/2eta$. It should include x and it still doesn't make sense if I just take the value into x.
4. I am not sure the assumption is correct that strongest adversary is near the radius. The malicious users could put a very big value datapoint outside the radius as well since it would be still updated by doing average.
5. The experiments are pretty weak. As claimed by the paper, the proposed method is a robust algorithm, it should include some malicious users in the simulation.

---

> ### Author Response · Authors · 2023-08-28
> **Response to Reviewer SJcN**
>
> Thank you for your valuable insights and suggestions. We have incorporated all of your feedback. We address each of your **Requested Changes** and **Cons** comments.
>
> **Requested Changes**
>
> 1. We now devote Section 2 to the failure-mode examples and defer all discussion of our method to Section 3. In Section 3.1.3, we briefly return to the examples in Section 2 to describe how our method successfully handles them. Hopefully this improves clarity.
>
> 2. We have added a description of Algorithm 1 in Section 2.1, directly after introducing it.
>
> 3. We have made all of these changes. Hopefully the figures are more readable and interpretable now.
>
> 4. We are currently running simulations with malicious users and will update as soon as we have the results. We also note that our *private label* experiment in Table 1, in which different clients assign different labels to the same MNIST image, is a type of malicious attack (also known as a *label flipping* attack in the Byzantine-robustness community), since a malicious client could try to corrupt the model by assigning the wrong label in training data to an image.
>
> **Cons**
>
> 1. See our response to **Requested Changes** 1. We now defer all discussion of our algorithm to Section 3 *after* the motivation has been established in Section 2.
>
> 2. In the revision we have edited the figures to make them more readable. Re Figure 1c), perhaps this is a misunderstanding. Figures 1a), b), and c) are examples we created to discuss the performance of *other* algorithms, not ours. The algorithms we discuss corresponding to each figure are: Myopic-Clustering for Figure 1a), IFCA and HypCluster from [G] and [M] respectively for Figure 1b), and Clustered Federated Learning from [S] for Figure 1c). We are claiming that our algorithm *Federated-Clustering* outperforms those other algorithms respectively on each of the examples depicted in the figures, not that Figure 1c) outperforms 1a) and 1b).
>
> 3. We got the values $1/2\eta$, $1/2\eta$, $-1/2\eta$ by evaluating $\nabla f_i(x)$ at the initialization point $x=1.5$. We have clarified this computation in the revision.
>
> 4. Any gradient which falls outside the ball is ignored, and instead assigned a value equal to the center of the ball (see line 4 of Algorithm 3). Therefore points outside the radius cannot move the center of the ball, and the strongest adversary is near the radius.
>
> 5. We are currently running experiments with malicious users and will put the results in the final revision. In Section 4.5, we have added simulations that verify that the number of thresholding rounds, $l$, scales with $\log{\sigma}$, as stated in Theorem 1. We are also running new experiments on CIFAR10 and CIFAR100 for the final revision.
>
> *References*
>
> [G] Ghosh, Avishek and Chung, Jichan and Yin, Dong and Ramchandran, Kannan. "An efficient framework for clustered federated learning". Advances in Neural Information Processing Systems, 33:19586–19597, 2020.
>
> [M] Mansour, Yishay and Mohri, Mehryar and Ro, Jae and Suresh, Ananda Theertha. "Three approaches for personaliza- tion with applications to federated learning". 2021.
>
> [S] Sattler, Felix and Muller, Klaus-Robert and Samek, Wojciech. "Clustered federated learning: Model-agnostic distributed multi-task optimization under privacy constraints". 2019.

---

> > ### Author Response · Authors · 2023-09-05
> > **Update on experiments**
> >
> > We have added two experiments with malicious workers in Section 4.4 showing that our algorithm is robust to such attacks. Additionally, Section 4.3 contains new experiments on CIFAR-10 and CIFAR-100 showing that our algorithm performs well and outperforms baselines. We hope these additions address the reviewer's concerns and please let us know if there are further questions.

---

### Author Response · Authors · 2023-08-29
**To All Reviewers**

We sincerely thank all reviewers for their detailed, helpful, and insightful suggestions which we incorporated. The feedback greatly improved our paper. All the edits to our resubmitted paper are in blue text, and all code is contained in the .zip file. We individually respond to each reviewer below.

---

### Decision · Action_Editors · 2023-10-04

**Recommendation:** Accept with minor revision

**Comment:**

This work achieves offers significant progress towards the theory of clustered federated learning, and manages to do so while offering robustness as well. The theory is sound, strong, and will likely be of significant interest to the community. The experiments help offer a variety of ablations and empirical validations of the theory, and have become relatively detailed since the initial review phase. This is a clear acceptance in terms of the content of the work.

Before submitting, I would like the authors to clean up their citations. In particular, at least one author name is misspelled (McMahon should be McMahan) and there are multiple instances where authors cite arXiv works instead of the published version of a paper, eg. "Personalized federated learning through local memorization" and "Very Deep Convolutional Networks for Large-Scale Image Recognition".

**Audience:**

The reviewers are also unanimous on this front (as am I). The paper establishes strong Byzantine robustness results for federated clustering, something that is of interest in multiple communities. Moreover, the paper is well-written (and has improved in organization and clarity due to the reviewer feedback), and the algorithm itself will likely attract attention from various members of the federated learning community.

**Claims And Evidence:**

The reviewers are unanimous (as am I) on the following points:

* The theoretical analysis is sound, and has matching (useful) lower bounds.
* The theoretical assumptions encompass a relatively broad range of settings (something that is not always true in work on Byzantine robustness)
* The experiments help demonstrate the empirical effectiveness of the methods (even though some reviewers asked for experiments on other datasets, which the authors have since added).

At least one reviewer believes that the paper is noteworthy for the thoroughness and impact of its theoretical results.

All in all this seems like the paper clearly supports its primary claims.